

# Continuous synchronization of the Greenland ice-core and U-Th timescales using probabilistic inversion

Francesco Muschitiello[1, 2] and Marco Antonio Aquino-Lopez[1]

[1]Department of Geography, University of Cambridge, Cambridge CB2 3EN, UK

[2]Centre for Climate Repair, DAMTP, Centre for Mathematical Sciences, Wilberforce Road, Cambridge, CB3 0WA, UK

*Correspondence to*: Francesco Muschitiello (fm476@cam.ac.uk)

**Abstract.** This study presents the first continuously measured transfer function that quantifies the age difference between the Greenland Ice-Core Chronology 2005 (GICC05) and the U-Th timescale during the last glacial period. The transfer function

was estimated using an automated algorithm for Bayesian inversion that allows inferring a continuous and objective synchronization between Greenland ice-core and East Asia Summer Monsoon speleothem data. The algorithm is based on an alignment model that considers prior knowledge on the GICC05 counting error, but also samples synchronization scenarios that exceed the differential dating uncertainty of the annual-layer count in ice cores, which are currently not detectable using conventional alignments techniques. The transfer function is on average 52% more precise than previous estimates and

significantly reduces the absolute dating uncertainty of the GICC05 back to 48 kyr ago. The results reveal that GICCC05 is, on average, systematically younger than the U-Th timescale by 0.97%. However, they also highlight that the annual-layer counting error is not strictly correlated over extended periods of time, and that within the coldest Greenland Stadials the differential dating uncertainty is likely underestimated by ~10-15%. Importantly, the analysis implies for the first time that during the Last Glacial Maximum GICC05 overcounts ice layers by ~15% –a bias attributable to a higher frequency of sub-

annual layers due to changes in the seasonal cycle of precipitation and mode of dust deposition to the Greenland Ice Sheet. The new timescale transfer function provides important constraints on the uncertainty surrounding the stratigraphic dating of the Greenland age-scale and enables an improved chronological integration of ice cores, U-Th-dated and radiocarbon-dated paleoclimate records on a common timeline. The transfer function is available as supplements to this study.

## 1 Introduction

The Greenland ice-core chronology 2005 (GICC05; Rasmussen et al., 2006; Svensson et al., 2008) and the U-Th timescale (e.g. Cheng et al., 2016) are among the most widely used independently dated chronological frameworks of the last glacial period. The timescales not only provide the backbone of some of the most unique and detailed records of global climate change, but their robustness make them exceptionally suited for resolving the temporal structure of Dansgaard-Oeschger (DO) events

and other abrupt climate shifts.





The GICC05 is based on annual-layer counting back to 60 kyr before 2000 AD (b2k) and due to the incremental nature of the counting uncertainty, it provides high internal consistency that enables accurate relative age estimates of climate events. The Greenland ice-core timescale underpins a number of high-resolution ice-core records of North Atlantic climate and atmospheric composition. These records have become established Northern Hemisphere templates for the last glacial period

and have shaped our understanding of the physical mechanisms driving rapid climate shifts (Andersen et al., 2004; Dahl-Jensen et al., 1998; Legrand and Mayewski, 1997; Schüpbach et al., 2018) and their rates of change (Jansen et al., 2020). By contrast, the U-Th timescale is constructed using high-precision U-Th dating, which yields much smaller uncertainty in the absolute ages than GICC05. The U-Th timescale provides a temporal framework for speleothem $\delta^{18}O$ data, which are largely dominated by records falling into the East Asian Summer Monsoon (EASM) domain (Corrick et al., 2020) (Fig. 1); altogether,

records from this region constitutes a key blueprint of low-latitude hydroclimate variability, integrating intensity changes in East Asian monsoon and meridional shifts of the intertropical convergence zone (ITCZ; Wang et al., 2006, 2001).

Both the GICC05 and U-Th timescale serve to test, improve and constrain chronologies for a wide range of paleoclimate archives and proxy records. These age scales have been used to validate and benchmark Antarctic ice-core chronologies (e.g. Buizert et al., 2015; Sigl et al., 2016), which ultimately enable resolving the inter-hemispheric phasing of DO events (Buizert

et al., 2018), and the rate of greenhouse gas emissions during the last glacial period (Bauska et al., 2021). They are also widely used to constrain paleoceanographic records with poor independent age control (Bard et al., 2013; Hughen and Heaton, 2020; Waelbroeck et al., 2019). To build a chronology for deep-sea sediment cores, proxy signals are commonly correlated to abrupt cooling and warming events observed in ice-core proxies or speleothem $\delta^{18}O$ under the assumption of direct synchrony of climate changes. These climatically tuned chronologies, despite limiting our ability to test leads and lags between oceanic and

atmospheric processes (Henry et al., 2016; Hughen and Heaton, 2020), still lay the foundations for deriving "best-guess" temporal constraints on a variety of fundamental boundary conditions of glacial ocean circulation and its coupling with the atmosphere system.

Because the Greenland ice-core and U-Th chronologies are constructed independently, the occurrence of systematic timescale offsets and dating biases of the order of hundreds of years complicates the comparisons of events integrated in the proxy

records that hinge on these timescales. Perhaps more importantly, the chronology that forms the older portion of the new IntCal20 radiocarbon calibration curve is dominantly reliant on the Hulu Cave speleothem U-Th timescale (Cheng et al., 2018, 2016; Reimer et al., 2020; Southon et al., 2012). During the period spanning ~14-54 kyr b2k, a wealth of $^{14}C$ datasets have been placed on the Hulu Cave U-Th timescale either indirectly via stratigraphic tuning of paleoclimate data to the high-resolution Hulu $\delta^{18}O$ record (Bard et al., 2013; Darfeuil et al., 2016; Hughen and Heaton, 2020), or more directly by means of

$^{14}C$ wiggle-matching (e.g. Bronk Ramsey et al., 2020; Turney et al., 2010, 2016). As a result, potential differences between the timescales –if not quantified and corrected for– can hinder a proper assessment of $^{14}C$-dated environmental and archaeological records within the ice-core climatic framework.



Furthermore, knowledge on the existing timescale offsets is important for high-resolution studies of marine $^{14}$C (e.g. Muschitiello et al., 2019). This is crucial for those reconstructions whose chronologies are more conveniently anchored to ice-core records rather than the Hulu Cave speleothems, as is often the case for North Atlantic sediment cores that integrate common regional climatic changes (Skinner et al., 2020; Thornalley et al., 2015; Waelbroeck et al., 2019) or sites where isochronous tephra deposits can be traced between ice cores and marine records (e.g. Ezat et al., 2017; Sadatzki et al., 2019; Skinner et al., 2020, 2017). In these instances, potential discrepancies between the GICC05 and U-Th timescales can lead to an imprecise assessment of ocean $^{14}$C concentrations relative to those of the atmosphere inferred from the IntCal datasets, thus affecting the estimation of ocean carbon inventories.

In turn, resolving the differences between the GICC05 and U-Th timescales can help to reduce and characterize their absolute dating uncertainty, and facilitate the comparison of ice cores and radiocarbon-dated records on a common timeline. Altogether, this is pivotal to advance our understanding of the physical mechanisms behind abrupt climate change, and to harmonize climate, environmental and archaeological records of the last glacial cycle.

There are two main types of synchronization to integrate the GICC05 and U-Th timescales: *i*) synchronization of climate records, and *ii*) synchronization of cosmogenic radionuclide data. The first is based on correlation of climatic signals integrated in Greenland ice-core records and speleothem $\delta^{18}$O that are assumed to be synchronous. The second is based on the correlation of externally-forced and essentially climate-independent variations in ice-core $^{10}$Be and Hulu Cave $^{14}$C records.

Despite the circularity that climate synchronization entails, such as precluding testing the synchronicity of teleconnections between Greenland and the East Asian monsoon system, the concerns about a potentially large climate phasing between polar ice-core and EASM speleothem records during the last glacial period have been put to rest. Unlike other regions where there are potentially complex site-specific responses to large-scale change (Adolphi et al., 2018), currently there is ample evidence that North Atlantic and Asian monsoon climate are coupled on short atmospheric timescales (e.g. Corrick et al., 2020; Cvijanovic et al., 2013) –i.e. likely shorter than the mean resolution of the proxy records used for climate synchronization. The teleconnection mechanism, which is likely modulated by variations in the Atlantic Meridional Overturning Circulation, is well documented and involves coherent meridional shifts of the mid-latitude storm tracks, the ITCZ, and the related monsoon systems (e.g. Ceppi et al., 2013; Kageyama et al., 2013; Zhang and Delworth, 2005). This is further supported by climate model simulations, which demonstrate that this atmospheric coupling synchronizes the North Atlantic and EASM region down to multidecadal timescales –a teleconnection that is robust under different glacial boundary conditions (Fig. 1c-d).

Since the construction of the GICC05 chronology, climate synchronizations between Greenland and EASM speleothem records (e.g. Hulu Cave) have been derived by identification of tie-points marking sharp transitions in both the ice-core and speleothem stratigraphies (e.g. DO events). The synchronization approach has been performed either by manual, qualitative comparison of the climate records (Corrick et al., 2020; Svensson et al., 2008, 2006; Weninger and Jöris, 2008), or using reproducible, quantitative methods for detecting change points (Adolphi et al., 2018; Buizert et al., 2014). To present, the main





methodological drawback of this approach is that it relies on only a discrete set of stratigraphic tie-points, which prevents quantifying the alignment uncertainties in a continuous fashion.

The approach for synchronizing cosmogenic radionuclide records involve sliding window techniques, such as cross-lagged regression (Muscheler et al., 2014) or more commonly Bayesian wiggle-matching (BWM; Adolphi and Muscheler, 2016). The techniques aim at matching relative changes in $^{10}Be$ and $^{14}C$ concentrations over a series of time windows, and by focusing

on centennial-to-multi-centennial variations that are typically dominated by solar-induced –and largely periodic– changes in production rates (Vonmoos et al., 2006). Synchronization using BWM offers high precision and has proved effective during the Holocene when the offsets between the ice-core and $^{14}C$ timescales are mostly systematic (Adolphi and Muscheler, 2016; Sigl et al., 2016). However, these matching techniques depend heavily on the predefined window length (e.g. Schoenherr et al., 2019) and can lead to biased conclusions about synchrony if the timescale offsets change faster than the time window used

for matching. Specifically, BWM only produces a single point of match for each window analysis, which generally spans 1,000 to 5,000 years, thus averaging out any short-term, nuanced fluctuations in the timescale difference (Adolphi et al., 2018; Adolphi and Muscheler, 2016). In addition, because of the highly autocorrelated nature of cosmogenic radionuclide records, BWM may misrepresent the relationship between the input $^{10}Be$ and $^{14}C$ signals. On one hand, autocorrelation can lead to neighbouring offset estimates of the BWM to be positively correlated, which results in smoothing out the timescale transfer

function. On the other hand, it may lead BWM to identify wrong correlation, thus yielding sudden –and potentially spurious– jumps in the timescale offsets (Muscheler et al., 2014; Muschitiello et al., 2019).

With climate synchronizations standing on a limited number of stratigraphic tie-point and the latest alignment of cosmogenic radionuclides on only five BWM estimates (Muscheler et al., 2020), a new continuous synchronization between the GICC05 and U-Th timescales for the last glacial period is urgently required. In particular, the recent revision of the high-resolution

Hulu Cave $\delta^{18}O$ record (Cheng et al., 2016; H. Cheng et al., 2021) with an updated U-Th chronology (Cheng et al., 2018; H. Cheng et al., 2021) provides further motivation for re-assessing the synchronization between the Greenland ice-core and U-Th timescales. Lastly, there is a need for improved constraints during the Last Glacial Maximum (LGM), i.e. when the timescales reach their largest offset, and assess possible fast changes in the timescale difference that are currently not detectable using wiggle-matching correlation techniques.

In this study, some of these limitations are addressed by applying an automated probabilistic synchronization method to produce the first continuous transfer function that quantifies the offset between the GICC05 and U-Th timescales. The method minimizes the misfit between ice-core and speleothem proxy records while accounting for prior knowledge on the uncertainty in annual layer identification in ice cores using a Bayesian inversion of the GICC05 maximum counting error (MCE) (Rasmussen et al., 2006; Svensson et al., 2008). To minimize noise due to site-specific environmental factors and U-Th dating

uncertainties, the speleothem records are integrated using a Monte Carlo Principal Component Analysis procedure that isolates the common EASM hydroclimatic pattern and estimates uncertainties. The new timescale transfer function considerably improve the precision and accuracy of earlier estimates and reduces the absolute dating uncertainty of GICC05 in the interval





~11-48 kyr b2k. The results also indicate large and fast fluctuations in the timescale difference during the LGM and other cold
stadial periods, suggesting previously unrecognised biases in the ice-core annual layer counting. The implications of these
findings are discussed.

## 2 Data and methods

### 2.1 Proxy data and Monte Carlo Principal Component Analysis

The offsets between the GICC05 and U-Th timescales were estimated performing a synchronization based on climate proxy
records (CLIM). The proxy data used in this study are presented in Fig. 2. The CLIM synchronization was established over
the period ~11-48 kyr b2k using a combination of high-resolution $Ca^{2+}$ concentrations of mineral dust from NGRIP on the
GICC05 timescale (Erhardt et al., 2019), and revised $\delta^{18}O$ data from EASM speleothems on their independent U-Th
chronologies (Corrick et al., 2020). As discussed above, the focus on speleothem records from the EASM domain is motivated
by *i*) the well-established in-phase climate coupling between the North Atlantic and the EASM system, *ii*) the overwhelmingly
large number of records from this region, *iii*) the key importance of the Hulu Cave U-Th chronology in the calibration of the
radiocarbon timescale.

Mineral dust aerosol in Greenland ice cores primarily originates from Asian deserts (Svensson et al., 2000). Its emissions are
strongly dependent on Asian hydroclimate via concerted shifts in the latitudinal position of the ITCZ and the EASM system
(Nagashima et al., 2011; Schiemann et al., 2009). Because NGRIP $Ca^{2+}$ indirectly register lower-latitude hydroclimate changes
mediated by latitudinal migrations of the ITCZ, it is suitable for direct comparison to EASM $\delta^{18}O$ data, which integrate ITCZ-
related shifts in monsoon rainfall over East Asia (Wang et al., 2006, 2001) with comparable durations (Fig. 3).

As for EASM speleothems, the proxy data are based on a compilation of 14 $\delta^{18}O$ records including the U-Th age determinations
underlying each speleothem chronology (Corrick et al., 2020) (Fig. 1a-b; Fig. 2b-c). The original compilation included 17
records and we removed 3 low-resolution and scarcely-dated records, i.e. whose median age resolution was less than 50 years
and had on average less than one U-Th age determination per thousand years. The data set from Hulu Cave was here updated
to incorporate recently published higher temporal resolution $\delta^{18}O$ measurements and additional U-Th dates (Cheng et al., 2016;
H. Cheng et al., 2021). To integrate all the $\delta^{18}O$ data into a single record representative of the EASM region, a Monte Carlo
Principal Component Analysis (MCPCA) was used (Fig. 2d). The method follows Anchukaitis and Tierney (2013) and is
presented and tested in detail therein. In brief, the procedure allows isolating the common large-scale pattern of hydroclimate
variability while accounting for age modelling uncertainties. The MCPCA method uses iterative age modelling of the available
U-Th ages and eigen-decomposition of the $\delta^{18}O$ records to produce a reduced set of orthogonal modes that reflect common
patterns of $\delta^{18}O$ variability and estimate uncertainties.

For this analysis 10,000 iterations of the MCPCA procedure were generated. In line with Anchukaitis and Tierney (2013),
each Monte Carlo iteration consists of the following steps: 1) the U-Th dates of each speleothem record are randomly resampled



within their probability distribution imposing that depositional ages increase monotonically with depth; 2) for each record an
age-depth model is fit to the resampled U-Th ages using a monotonic piecewise cubic hermite spline; 3)  the leading PCA
mode (PC1) of the 14 $\delta^{18}$O proxy records is calculated using probabilistic PCA (Tipping and Bishop, 1999) and ensuring that
the sign of the eigenvector is consistent across iterations. The resulting 10,000 ensemble members of the PC1  were used to
estimate median and confidence levels, which were ultimately employed as a target record for the synchronization method
described below (hereafter referred to as EASM PC1).

**2.2.1 Probabilistic algorithm for proxy-data synchronization**

As discussed in Section 1, tie-point and wiggle-matching synchronizations have inherent problems that limit estimating the
alignment of proxy timeseries in a continuous fashion. The alignment uncertainty of tie-point synchronizations is poorly
characterized, tie points can be difficult to reproduce, and even when they are defined statistically the synchronization still
does not account for potential shared signal structures in between consecutive ties.

Probabilistic alignment methods have a unique and underexploited potential to correlate proxy timeseries and move away from
point-wise and wiggle-matching synchronization techniques. They are especially well suited for establishing continuous
alignments and can help matching previously untapped common structures in the signal of climate and cosmogenic
radionuclide records. These methods are fully automated and have the advantage of ensuring reproducibility, deriving credible
bands associated with the alignment process, and inferring the probability of synchronization solutions based on prior
constraints on accumulation rates (e.g. Lin et al., 2014; Muschitiello et al., 2020; Parrenin et al., 2015).

In this study, a continuous synchronization of the GICC05 to the U-Th timescale is established using an appositely developed
automated algorithm for probabilistic inversion. The inverse problem is formulated using a Bayesian framework in order to
sample the full range of possible GICC05–U-Th synchronization scenarios and explicitly build in prior ice-core chronological
constraints. Assuming that the U-Th timescale is absolute, our inverse scheme simulates the age offset history between GICC05
and the EASM PC1 (i.e. see Section 2.1) in response to changes in Greenland ice accumulation over time. The link requires a
likelihood function, which quantifies how probable the alignment between ice-core and speleothem records is given a particular
simulated ice-core depositional history.

The numerical approach builds upon previous work using a hidden Markov model for automated synchronization of
paleoclimate records (Cutmore et al., 2021; Muschitiello et al., 2020, 2019; Sessford et al., 2019; West et al., 2021, 2019). The
model employed here uses constraints imposed by the MCE, i.e. the accumulated absolute annual layer counting error of the
Greenland ice-core chronology, to deform the entirety of an input timeseries (on the GICC05 timescale) onto a target (on the
U-Th timescale). The method minimizes the misfit between the input and the target, and finds a sample of alignments between
Greenland ice cores and the EASM PC1 record that are physically coherent with the absolute dating uncertainty of GICC05
and some of its counting error properties. However, it should be born in mind that the model used in this study does not provide
a fully comprehensive representation of the complexity that characterizes the ice-core layer counting procedure and its





uncertainty (Rasmussen et al., 2006; Svensson et al., 2006). Rather, our approach aims at approximating the counting structure of GICC05 in order to infer estimates of the synchronization uncertainty. The method is also adaptable to a variety of formulations of the inverse problem and to using multiple input and target records simultaneously when determining the alignment.

### 2.2.2 Inverse modelling approach

In order to establish a continuous alignment between the GICC05 and U-Th timescale, we need to define an alignment function, which relates the GICC05 age of the input NGRIP $Ca^{2+}$ record to unknown U-Th ages associated with the target EASM PC1 record. The function is here defined by a mathematical representation of the GICC05–U-Th age relationship that effectively allows linearly stretching and compressing the ice-core chronology relative to the U-Th timescale. To estimate this alignment function, we propose a piece-wise linear function ($\tau(\cdot)$) with $K$ equally spaced sections within the domain, where the slopes at each section are $m_i > 0$. This ensures no time reversals by guaranteeing that the function is monotonically increasing. The function is also influenced by the initial shift of the alignment. If $\tau(\cdot)$ represents the alignment function, this initial shift can be defined as $\tau_0 = \tau(t_0)$. Hence, the parameters of $\tau(\cdot)$ are defined as $(\tau_0, m)$, where $m$ is a $K$-dimensional vector containing the slopes of each evenly spaced section. The model can then be expressed as:

$$\tau(t) = \tau_0 + \sum_{i=1}^{j} (m_j \Delta c) + m_{i+1}(t - c_i), \tag{1}$$

where $c_i \leq t < c_{i+1}$, $0 \leq i < K$, and $c_0 < c_1 < \cdots < c_K$ represent evenly spaced time intervals along the GICC05 timeline with length $\Delta c$. The slopes $m = (m_1, m_2, \ldots, m_K)$ correspond to the linear sections within each interval. This function is analogous to the one employed to construct age-depth models (Blaauw and Christen, 2011), i.e. a piece-wise linear function with positive slopes, which prevents time-reversals but preserves the shifts ($m_j \Delta c$) from younger to older sections.

However, our proposed method for calculating the alignment function, $\tau(t)$, significantly differ from the one presented by Blaauw and Christen (2011), which employs an autoregressive gamma process for simulating sedimentation rates with downcore dependence. Unlike their method, ours restricts such information exchange, thereby preserving the independence of each section. This feature is advantageous given our data context, which underlie very distinct depositional environments (i.e. ice cores versus speleothems) and different proxy measurements. It is important to note that since the input and target records are provided on their own independent chronologies, we do not need to model the autocorrelation associated with each timescale. Therefore, our approach, which maintains the independence of each section, is better suited for these datasets.

Our task is the development of a method for inferring the alignment function $\tau(t)$, which minimizes the misfit between NGRIP $Ca^{2+}$ on the GICC05 timescale $\vec{t'}$, and the EASM PC1 on the U-Th time scale $\vec{t}$. Let's define the $i^{th}$ data point in the input ice-core record associated with time $t'_i$ on the GICC05 timescale as $g_i$. Therefore, the vector $\vec{g}$ denotes the input signal, containing





the NGRIP Ca$^{2+}$ measurements on the GICC05 timescale. Analogously, for the EASM PC1 record, each data point, defined as $u_j$, represents the target signal at time $t_j$ on the U-Th time scale. The vector $\vec{u}$ houses this target signal and contains the data from the EASM PC1 record on the U-Th timescale.

With this notation, we regard each datum in $\vec{u}$ and $\vec{g}$ as an observation of the proxy at a specific point in time $(t_i)$, following a normal distribution where the mean corresponds to the "true" value of the proxy at time $t_i$ ($\vec{U}$ and $\vec{G}$). In this case, the input

record is reported without errors, so it can be assumed that $\vec{G} = \vec{g}$. On the other hand, the target record is accompanied by uncertainty $(\sigma^2_{u_i})$, so we assume $u_i \sim \mathcal{N}(U_i, \sigma^2_{u_i})$, where $\sigma^2_{u_i}$ denotes the variance of each data point $u_i$. It should be noted that $u_j$ is associated with a time $t_i$, thus the appropriate notation is $u_j(t_i)$. Assuming that both input and target are adequately synchronized by the alignment function $\tau(t)$, $u(t_i) \approx g(\tau(t_i)) \, \forall \, t_i \in \vec{t}$ holds true. To calculate the goodness of the fit between $\vec{u}$ and $\vec{g}$, and given that the uncertainty associated with $\vec{u}$ can be considered a random variable, we here use the

approach devised by Christen and Pérez (2009). We therefore define $u_i$ as follows:

$$u_i \mid \tau_0, m, t_i, \vec{G}, \sigma^2_{u_i} \sim t\big(G(\tau(t_i)), \sigma^2_{u_i}, a, b\big), \tag{2}$$

where $a$ and $b$ are the parameters of the t-distribution (Christen and Pérez , 2009). It is important to stress that $G$ is set only at discrete times, $\vec{t'}$, which may not necessarily match an observed U-Th age on the target EASM PC1 record ($\vec{t}$). To obtain values for $G(\tau(t_i))$ at any specific time, we employ linear interpolation using the observations from GICC05. This method

enables us to compute $G(t)$ for any given $t \in (t'_0, t'_m)$, where $(t'_0, t'_m)$ is the time window of the input ice-core record on the GICC05 timescale, whereby $t'_0 = 10.75$ kyr b2k and $t'_m = 48$ kyr b2k.

**2.2.3 Model's likelihood, parameters and priors**

Since the synchronization process is fundamentally uncertain, we apply a Bayesian approach to infer the optimal alignment between $\vec{u}$ and $\vec{g}$ that accounts for glaciological information on unobserved changes in Greenland ice accumulation. For the

correct implementation of this approach we require a likelihood function. This likelihood function evaluates the previously mentioned assumption ($U(t_i) \approx G(\tau(t_i))$) of the aligned ice-core record and the target speleothem data, given a particular set of parameter values $\Phi = (\tau_0, m, \vec{G}, \sigma^2_{u_i})$. The model is defined by Eq. 3, which allow us to calculate the log-likelihood of the data as:

$$\ell \propto \sum_{i=0}^{n} -\log(\sigma_{u_i}) - \frac{a}{2} log\left(b + \frac{G(\tau(t_i)) - u_i}{2\sigma_{u_i}}\right)^2, \tag{3}$$

In the synchronization problem posed here, the likelihood function determines the goodness of the fit by minimizing the mismatch between the input and the target at every data point on the U-Th timescale, i.e. by optimizing the function $\tau(\cdot)$. $\tau_0$ and $m$ are model parameters, whereas $\vec{G}$, $\sigma^2_{u_i}$, and $t_i$ are related to the data. $\vec{G}$ refers to the input data on the GICC05 timescale,





and $\sigma_{u_i}^2$ refers to the reported variance of the target record on the U-Th timescale. Any underestimation is addressed by using the t-distribution (Christen and Pérez , 2009). Lastly, $t_i$ represents the time of the i-th target signal on the U-Th timescale.

The model estimates the probability of a given alignment that relates GICC05 and U-Th ages by evaluating the log-likelihood function described in Eq. 3 together with prior distributions for $\tau_0$ and $m$. To avoid sampling outside a physically reasonable range and to identify a sample of optimal synchronizations, this probability is based on prior knowledge that the alignment between the input and target is largely limited by the constraints imposed by the absolute counting error of GICC05. Given that the parameters $\tau_0$ and $m$ are largely unknown and must be inferred from the model, we assign uninformative priors to

ensure that the posterior alignment is majorly influenced by the data. For $\tau_0$, which defines the initial timescale offset at time $t_0$, we apply a prior distribution of $\tau_0 \sim \mathbb{1}_{(0<m_i<5\ \mathrm{RCE})}\mathcal{N}(t_0, \sigma_0)$, where $\sigma_0$ is the Relative Counting Error (RCE) at $t_0$, in our case $\sigma_0 = RCE_{t_0}$, which is defined as the rate of change of the MCE. The decision to use RCE instead of MCE is justified because RCE integrates the speed at which the counting error changes. This makes RCE a more suitable metric for constraining both the initial shift and the slopes of the function $\tau$.

Since one of the objectives of this study is to invert changes in the GICC05–U-Th timescale difference (hereafter expressed as $\Delta T = t' - t$) brought about by possible unaccounted biases in the ice-core annual layer counting, the prior knowledge should allow sampling a RCE greater than the nominal values imposed by the GICC05, which during the last glacial period range ~5-10% (Svensson et al., 2008, 2006). In our model, each slope within the vector $m$ is independent and strictly positive. The slopes reflect the rate at which the age relationship between the GICC05 and U-Th timescales changes over two adjacent

sections, whereby $m_i > 1$ implies a linear stretching of GICC05, and $0 > m_i > 1$ implies a compression. With this understanding, we assigned $m_i$ a truncated normal distribution $m_i \sim \mathbb{1}_{(0<m_i<5\ \mathrm{RCE})}\mathcal{N}(0, \sigma_m^2)$, which ensures that $m_i$ remains positive. This effectively allows the model exploring RCE values up to 5 times greater than nominal and considering GICC05–U-Th synchronizations that exceed the range allowed by the MCE (as is generally the case for the Holocene, e.g. Adolphi and Muscheler, 2016), as well as accommodating propagation of the age errors associated with the U-Th timescale. To ensure that

$m_i$ follows the GICC05 counting error, $\sigma_m^2$ is fixed to be as half the RCE at a given time $t$. In our implementation, we have set the value of $K$ to 200, which implies that the timespan of $U(t_i)$ is divided into 200 sections (i.e. ~180 years per section). This division provides a reasonable compromise between computational performance and alignment resolution, ensuring that each section contains enough data for meaningful results.

In order to calculate a posterior sample of the parameters and optimizing the function $\tau(\cdot)$, a Markov chain Monte Carlo

methodology is used (see Section 2.2.4). This approach not only aims to find the best parameter values but also generates samples of the posterior distributions for each parameter. These samples are valuable for inferring uncertainties related to the alignment and can even be utilized to propagate the latter into the aligned proxies.

### 2.2.4 MCMC: determining the posterior distribution

Because of the nonlinear nature of the synchronization problem and the fact that there are far too many alignments to calculate
all their probabilities, a stochastic Monte Carlo method is required to explore the posterior distribution in a computationally
efficient way. Calculation of the posterior probability proceeds by sampling an initial value for each unknown model parameter
from the associated prior distributions using Markov chain Monte Carlo (MCMC). MCMC techniques play a key role in
statistical analysis, providing a systematic method for sampling from complex, multidimensional posterior distributions. The
principles of MCMC algorithms are based on the concept of a Markov chain, where the future state solely depends on the
current state, not on the series of previous states. Beginning from an arbitrary point, the MCMC algorithm initiates a sequence
of steps or "leaps" across the parameter space. The direction and magnitude of each leap are governed by a set of predefined
rules specific to each MCMC method. Over time, this sequence of leaps effectively samples the target distribution. Regardless
of where it starts, the algorithm ensures that it will converge to and accurately sample from the target distribution, provided it
completes enough iterations (Brooks et al., 2011). The time or iterations required for the algorithm to stabilize and accurately
reflect the posterior distribution is commonly referred to as the "burn-in" period. This convergence is what allows us to obtain
accurate samples from the posterior distribution, enabling us to infer the model's parameters.

In this study, MCMC is driven by a Differential Evolution Markov Chain (DE-MCz) sampler (Ter Braak and Vrugt, 2008),
which is particularly effective in dealing with multi-modal posterior probability distributions. The DE-MCZ method is
designed to update various segments of the Markov chain simultaneously, which significantly speed up the processing time
for large multidimensional datasets. Additionally, DE-MCz's inherent adaptability makes it an excellent choice for tackling
complex problems, and an ideal MCMC for our implementation.

The sampler was ran for $1.5 \times 10^6$ MCMC iterations after disregarding a burn-in time of $0.5 \times 10^6$ steps and only retaining
every $10^{th}$ iteration to mitigate the statistical dependence of the model parameters. This was deemed to be a sufficiently long
MCMC run for the simulation to reach convergence, as monitored by a multivariate potential scale reduction factor less than
1.1 (Brooks and Gelman, 1998). The sample from the remaining iterations was used to estimate the posterior distribution of
each random variable in the model ($\tau_0$ and $m$). By leveraging samples from these parameters, we computed a posterior sample
of alignment functions $\tau(\cdot)$. This posterior sample of functions then allows us to evaluate metrics such as the median and
credibility intervals. However, note that this is simplification of the process behind it: because the resulting output is a sample
of random variables, the most appropriate way to report these results is the ensemble of samples from $\tau(\cdot)$. Nevertheless, in
order to simplify the output and follow common practice, we reported the median and 95% credible intervals.

## 3 Results and discussion

### 3.1 Synchronization and timescale transfer function

The leading mode of the MCPCA procedure –i.e. the EASM PC1– provides the target record for the CLIM synchronization
and is presented in Figure 2d. The PC1 is dominated by the characteristic EASM hydroclimate signal, and even though the





Monte Carlo approach results in some temporal smoothing, the millennial-scale trends and shorter events that punctuated the last glacial period in this region are reasonably well captured. The Hulu Cave record has the largest loading on PC1 as it is the data set with the highest temporal resolution and the only one stretching over the whole synchronization interval. It loads highly especially between ~16-28 kyr b2k where there are fewer speleothem records.

The inverted CLIM synchronization history and related GICC05–U-Th timescale transfer function are presented in Figure 4. Assuming that the U-Th timescale is accurate and considering the age uncertainties associated with EASM PC1, it can be observed that throughout the last glacial period the timescale difference $\Delta T$ is well within the MCE limits of GICC05. Notably, GICC05 is systematically younger than the U-Th timescale and the age difference increases with time, indicating, on average, a stretch of 0.97% during the construction of GICC05, which is comparable to the 0.63% linear scaling bias estimated by Buizert et al. (2014) (Fig. 4c).

Nonetheless superimposed upon this upward trend there are a number of shorter-term $\Delta T$ fluctuations. The CLIM results indicate that at the onset of the Holocene GICC05 is slightly older than the U-Th timescale, yielding a $\Delta T$ of $-45^{-25}_{-65}$ years (95% credible range) (Fig. 4c). Between ~11 and 15 kyr b2k, GICC05 is gradually stretched and becomes younger than the U-Th timescale, resulting in $\Delta T$ values of $+10^{+95}_{-70}$ years. In the interval ~15-24 kyr b2k, corresponding to the early stage of GS-2, GICC05 is further stretched but slightly faster than allowed by the annual counting error, reaching a maximum $\Delta T$ of $+450^{+550}_{+355}$ years, which is as large or a little larger than the MCE permits. From ~24 to 26 kyr b2k, which corresponds to GS-3 and approximately in phase with the global LGM ice-volume peak (Hughes and Gibbard, 2015), again the GICC05 annual-layer count changes faster than the counting error allows, highlighting a possible compression of the timescale with $\Delta T$ values dropping to $+160^{+250}_{+55}$ years. Around ~28 kyr b2k, another short-term stretch of GICC05 is observed, when $\Delta T$ values raise to $+320^{+420}_{+210}$ years. Before ~30 kyr b2k, $\Delta T$ exhibits –within the uncertainty bounds– stable and gradually increasing values reaching $+430^{+600}_{+250}$ years at 48 kyr b2k.

The largest $\Delta T$ excursions between ~15 and 27 kyr b2k are primarily controlled by the millennial-scale and shorter-term variability recorded in NGRIP $Ca^{2+}$ and EASM PC1 (Fig. 4a-b). The overall match is driven by the alignment between the GS-3 dust peaks in the Greenland ice cores (Rasmussen et al., 2008) and the well-defined declines in monsoon strength observed in EASM PC1 (Fig. 4a), as well as by a few marked proxy excursion during GS-2 (Fig. 4b). However, the transfer function error is large during GS-2. This is mainly due to the relatively lower signal-to-noise ratio in the climate records across the Lateglacial and LGM, where the alignment is less robust. In the case of the LGM, further work and independent age constraints would therefore be desirable to improve the match between the GICC05 and U-Th timescales. Before ~28 kyr b2k, the $\Delta T$ results are constrained by the match between stadials-interstadial transitions in NGRIP $Ca^{2+}$ and the corresponding monsoon events integrated in the EASM PC1 (Fig. 4a), with the exception of the interval surrounding GS-10 and GI-10 at ~42 kyr b2k when the speleothem $\delta^{18}O$ data underpinning the EASM PC1 exhibit some temporal inconsistencies, thus resulting in a larger alignment error.



A comparison of the CLIM timescale transfer function alongside published $\Delta T$ estimates is presented in Figure 5. The inferred timescale offset history is in good agreement with independent $\Delta T$ estimates based on BWM of cosmogenic radionuclide records, match points between GICC05 and the $^{14}$C timescale, and stratigraphic matching using atmospheric methane in ice

cores (Martin et al., 2023). The uncertainty bounds of the new transfer function is overall ~52% narrower than previous estimates based on BWM (Adolphi et al., 2018; Muscheler et al., 2020). Uncertainties are on average ~30% smaller during deglaciation and LGM (i.e. after 28 kyr b2k), whereas the precision is improved by up to ~73% during Marine Isotope Stage 3 (i.e. prior to 28 kyr b2k).

Not only the new timescale transfer function is considerably more precise, but the continuous synchronization brings to light

a more complex GICC05–U-Th age difference history than previously assumed. Moreover, our synchronization model allowed identifying a number of potential fast changes in the timescale difference –i.e. features that would have gone undetected had the model been more tightly constrained by the nominal RCE of GICC05.

### 3.2 Differential dating uncertainty of GICC05

Assuming that the U-Th timescale is absolute, a new picture emerges showing that the identification of uncertain annual layers

in the GICC05 is potentially less accurate than previously thought (Fig. 5-6). The GICC05 timescale appears to be either missing or gaining time beyond its RCE during some of the longest and coldest stadials (Fig. 6). Too few annual layers have been identified within H1/GS-2 and GS-4, and less markedly during H3/GS-5 and GS-6, whereas too many layers have been counted over H2/GS-3, i.e. the LGM. This in principles challenges the layer counting method and uncertainty estimates and implies that the bias in the GICC05 layer counting is not systematically depending on accumulation rates.

The observation that GICC05 likely undercounts ice layers during H1/GS-2 is quantitatively comparable to previous results (Adolphi et al., 2018) and in line with results from methane synchronization between WIAS Divide and GISP2 (Martin et al., 2023) (Fig. 5). Specifically, CLIM highlights that GICC05 counts on average ~10% too few annual layers in the interval ~15-18 kyr b2k. Similarly, during GS-4, which represents the coldest period recorded in Greenland ice cores (Fig. 6a), we observe an undercount of ~15% centred at ~28ky b2k.

Perhaps a more interesting result is that throughout most of LGM/GS-3 there is an increasing tendency to count too many years in the GICC05 stratigraphy. The overcount starts at ~24 kyr b2k and reaches a maximum rate of change at ~26 kyr b2k, when the GICC05 timescale counts on average ~15% too many layers (Fig. 6c). The LGM/GS-3 interval is one of the coldest section of the last glacial period (Fig. 6a-b) and an overcount of annual layers is seemingly at odds with the general assumption that fewer years have been detected during stadials, i.e. when low accumulation rates and thinner ice layers make the

identification of annual layers more difficult (Rasmussen et al., 2006; Svensson et al., 2008, 2006). However, a bias towards counting too many ice years during LGM/GS-3 is not unexpected as there are some weak indications that additional annual layers have been counted in other cold sections of the GICC05 stratigraphy (Andersen et al., 2006; Rasmussen et al., 2006; Svensson et al., 2006).



This finding requires further consideration. During LGM/GS-3, the RCE inferred from CLIM maps onto the dust concentration
profile in Greenland ice cores (Fig. 6b-c). This interval features the two most distinct and pronounced dust peaks of the last
glacial period, in which dust levels increase by a factor of 3 in NGRIP ice cores (Ruth et al., 2003). Since high dust content in
the ice is notoriously liable to complicate the annual-layer counting in a number of ways, this correspondence suggests a
possible impact of dust deposition on the identification of the annual layers.

The layer counting in the coldest climatic events of the GICC05 stratigraphy relies mostly on the visual identification of annual
variations in two parameters over the NGRIP ice cores. Since the chemical records do not resolve the thin stadial layers,
counting is constrained using the high-resolution visual stratigraphy (VS) grey-scale refraction profile (Svensson et al., 2005)
and the electrical conductivity measurement (ECM) on the solid ice (Dahl-Jensen et al., 2002; Hammer, 1980). The VS profile
represents the depositional history at NGRIP. Inspections of the VS data throughout the glacial period highlights a strong
correlation between the frequency of visible layers and dust concentration, suggesting that the intensity (i.e. the grey value) of
each layer is related to its impurity content representing an individual dust depositional event (Svensson et al., 2005). This
may results in the VS record to contain multiple visible large layers per year, which can complicate the counting and lead to a
misinterpretation of the actual annual signal. On the other hand, the ECM is strongly dominated by variations in dust (e.g.
Taylor et al., 1997). The ECM profile is attenuated in sections with high concentrations of dust due to the increased alkalinity,
thus subduing the annual cycle in the ECM signal and making the resolution of this parameter marginal for the identification
of annual layers (Andersen et al., 2006; Rasmussen et al., 2008). For these reasons, greater dust deposition rates may limit the
use of the VS and ECM data for direct counting of annual layers.

Moreover, accurate counting over the cold sections that feature multiple depositional events depends on the untested
assumption that clusters of peaks in the VS and ECM profiles reflect seasonal variations in dust deposition resembling those
observed in the shallower parts of the NGRIP ice core. Modern dust emissions from Asian deserts peak in the Northern
Hemisphere spring. This peak is generally associated with enhanced flux of dust to the ice (Beer et al., 1991; Bory et al., 2002;
Whitlow et al., 1992) –a signature consistent with that of the warmest sections of the GICC05 stratigraphy and coinciding with
layers of high refraction in the VS record (Ram and Koenig, 1997; Rasmussen et al., 2006; Svensson et al., 2008). However,
an altered atmospheric circulation and precipitation pattern during the LGM may have caused fundamental changes in the
seasonality, magnitude, frequency and mode of deposition of dust impurities to the Greenland Ice Sheet.

Model simulations of the dust cycle under glacial climate conditions show a prolongation of the dust-emission season with a
two- to three-fold increase in atmospheric emissions and deposition rates in the high northern latitudes during the LGM
compared to modern (Werner et al., 2002). The dominant factor driving the higher dust emission fluxes at the LGM appears
to be increased strength and variability of glacial winds over the dust source regions. Evidence from general circulation models
(Kageyama and Valdes, 2000; Li and Battisti, 2008; Löfverström et al., 2016; Ullman et al., 2014) and proxy reconstructions
(L. Cheng et al., 2021; Kageyama et al., 2006; Luetscher et al., 2015; Spötl et al., 2021) consistently point towards stronger
northern westerlies at the LGM. In particular, GS-3 stands out in the context of the last glacial period as the phase when this





altered flow pattern was most extreme (Fig. 6d), i.e. in conjunction with the maximum extent of the Laurentide Ice Sheet, which caused a strengthening and southward deflection of the westerlies (e.g. Löfverström and Lora, 2017).

Increased emissions and transport to Greenland provide an explanation for the high dust concentrations in the ice observed
during LGM/GS-3. However, to justify that GICC05 contains 15% too many annual layers during GS-3, additional factors have to be invoked. For example, changes in the seasonality and mode of precipitation can play a key role in increasing the number of dust depositional events that ultimately control the frequency of sub-annual layers observed in the VS profile. Model simulations suggest that at LGM Greenland experienced a marked reduction in winter and spring precipitation and a shift to a precipitation regime with a pronounced summer maximum –in contrast to the characteristic modern springtime peak (Krinner
et al., 1997; Merz et al., 2013; Werner et al., 2002). It has been shown that lower precipitation rates and a shift in seasonality of precipitation inhibit wet deposition of dust during glacial winter and spring. This leads to a substantial increase in the contribution of dry deposition processes at Summit, which produce dust spikes in ice cores that are less evenly distributed over depth than modern ones (Werner et al., 2002). Dry deposition commonly occurs through gravitational settling and turbulent redistribution of snow to the surface and is thus more conducive to increasing the frequency of annual dust depositional events
registered in ice-core records. Hence, the increased seasonality of LGM precipitation and enhanced dry deposition over Greenland may explain the higher frequency of sub-annual layers in the VS signal and the resulting overcount of annual layers in GICC05 during GS-3. A complete understanding of the physical processes that led to the overcount during GS-3 is however beyond the scope of this work and requires more detailed investigations using climate model experiments of dust transport and deposition.

**4 Conclusions**

The first continuous climate synchronization between the Greenland ice-core chronology 2005 (GICC05) and the U-Th timescale is presented. The synchronization was established using an automated alignment algorithm for Bayesian inversion of the annual-layer counting uncertainty of GICC05. The algorithm quantifies the probability of alignments between Greenland ice-core and East Asia summer monsoon speleothem signals, and infers the age difference between the underlying timescales.
The synchronization method evaluates possible shifts in the timescale difference that exceed the differential dating uncertainty of GICC05, which are not easily quantifiable using traditional tie-point correlation or wiggle-matching techniques.

The new synchronization results are consistent with independent reconstructions and improve the average precision of the GICC05–U-Th timescale transfer function by ~52% relative to previous estimates. Based on the assumed accuracy of the U-Th timescale, the results significantly reduce the absolute dating uncertainty of the Greenland timescale back to 48 kyr b2k
and indicate that the MCE is generally a conservative uncertainty measurement.

Yet, the analysis shows that the relationship between the GICC05 and the U-Th timescale is potentially more variable than previously assumed and that the annual-layer counting error of the ice-core chronology is not necessarily correlated over long

periods of time. It is found that within the coldest stadials, GICC05 is either missing or gaining time faster than allowed by its nominal differential dating uncertainty. The annual-layer count identifies on average ~10-15% too few ice years within H1/GS-
2 and GS-4. In contrast ~15% too many ice years may have been counted within GS-3, i.e. in conjunction with the LGM.

The results imply a major shift in the differential counting within the interval ~24-26 kyr b2k, when the difference between the GICC05 and the U-Th timescale drifts from +450 to +160 years. The reason for this marked overcount is attributed to a misinterpretation of the annual-layer record over GS-3. This is likely due to an increased occurrence of multiple-layer years resulting from a higher frequency of dust depositional events at the LGM in response to changes in seasonality of precipitation
and a greater contribution of dry deposition processes. This is an important point, as a large counting bias within GS-3 may explain why it has been difficult to identify a robust bipolar volcanic match between Greenland and Antarctic ice cores during LGM (Svensson et al., 2020).

This study illustrates the utility of probabilistic inversion methods to infer continuous and objective synchronizations of paleoclimate records. The new timescale transfer function presented here sets important constraints on the biases that
accompany the stratigraphic dating of GICC05 and will facilitate the comparison of ice cores, U-Th-dated and radiocarbon-dated records on a common timeline.

**Data availability.** The stack of speleothem $\delta^{18}$O records (EASM PC1) presented in Figure 2 and the CLIM transfer function presented in Figure 4 are available as supplements to this paper.

**Code availability.** All R code used for synchronization analysis is available from the corresponding author upon request.

**Competing interests.** The authors declare no conflict of interest.

**Acknowledgements.** The authors acknowledge funding from a Natural Environment research Council (NERC) Discovery Science Grant (NE/W006243/1), and the support from the Isaac Newton Trust at the University of Cambridge (LCAG/444.G101121). This study is a contribution to the INTIMATE (INTegration of Ice-core, Marine, and Terrestrial records) project.



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



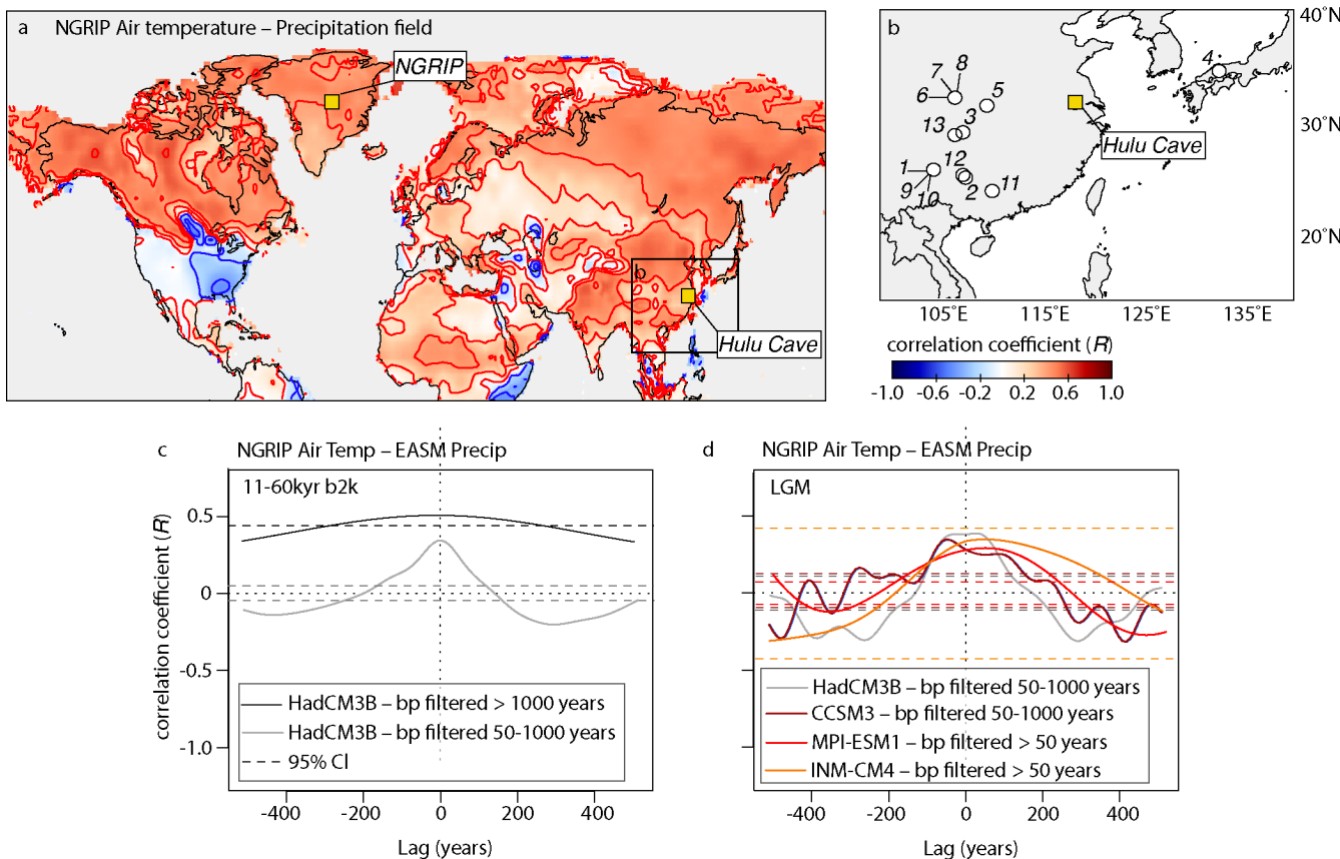

**Figure 1. a.** Instantaneous (lag 0) spatial correlation between mean decadal surface air temperature at the NGRIP site and mean decadal land

surface temperatures during the last glacial period (11-60 kyr b2k) as simulated with the HadCM3B-M2.1 coupled general circulation model
(Armstrong et al., 2019). The simulation incorporates Dansgaard-Oeschger cycles, Heinrich Events, and shorter-term variability, with a
spatial climate fingerprint derived from a Last Glacial Maximum (LGM) freshwater hosing experiment applied over the North Atlantic
Ocean. The location of NGRIP, the East Asian summer monsoon (EASM) region and Hulu Cave are also shown. **b.** EASM domain showing
the location of speleothem records compiled by Corrick et al. (2020) and used in this study. Numbering: 1. Dashibao; 2. Dongge; 3. Furong;

4. Maboroshi; 5. Sanbao; 6. Shizi; 7. Songjia1; 8. Songjia3; 9. Wulu3; 10. Wulu32; 11.Xiaobailong; 12. Yamen; 13. Yangkou. Reference to
the cave site is provided in Corrick et al. (2020). **c.** Cross-correlation between simulated NGRIP air temperature and EASM precipitation
between 11 and 60 kyr ago. The EASM region is defined here as the average of 10-40˚N and 95-125˚E. The timeseries were bandpass filtered
to quantify leads and lags at millennial and shorter timescales, respectively. **d.** Same as (c) but for the LGM using transient climate model
simulations (Armstrong et al., 2019; Liu et al., 2009) and equilibrium experiments from CMIP6 (Kageyama et al., 2021). Results from

HadCM3B and CCSM3 span the interval ~17.5-21kyr b2k, whereas for CMIP6 only simulations longer than 200 years were considered for
the cross-correlation analysis. Dashed lines reflect the 95% significance level against first order autoregressive (AR1) noise.







**Figure 2.** Proxy-climate data used for the climate synchronization (CLIM) presented in this study and shown on their original timescale. **a.** Mineral-dust derived $Ca^{2+}$ ion concentration record from NGRIP (Erhardt et al., 2019) on the GICC05 timescale (Rasmussen et al., 2006;

Svensson et al., 2008). Partitioning of Greenland Interstadials (GI) and Greenland Stadial (GS) 2 are indicated. **b.** High-resolution Hulu Cave $\delta^{18}O$ record (Cheng et al., 2016; H. Cheng et al., 2021) on the revised U-Th timescale (Cheng et al., 2018; H. Cheng et al., 2021) and individual U-Th measurements with their $\pm 2\sigma$ uncertainty (grey squares). **c.** Stack of speleothem $\delta^{18}O$ records from the East Asian Summer Monsoon (EASM) region presented in Corrick et al. (2020) and used in this study. $\delta^{18}O$ values are expressed as anomalies from the record mean. Individual U-Th measurements associated with each record are also presented. Numbering refers to location of cave sites shown in

Fig. 1b. **d.** First principal component (PC1) of the 14 EASM speleothem records presented in (c) from the MCPCA procedure used in this study (see Section 2.1 for details). The solid line indicates the median from the 10,000 member ensemble, while shadings reflect the empirical 68 and 95% confidence intervals from the ensemble.





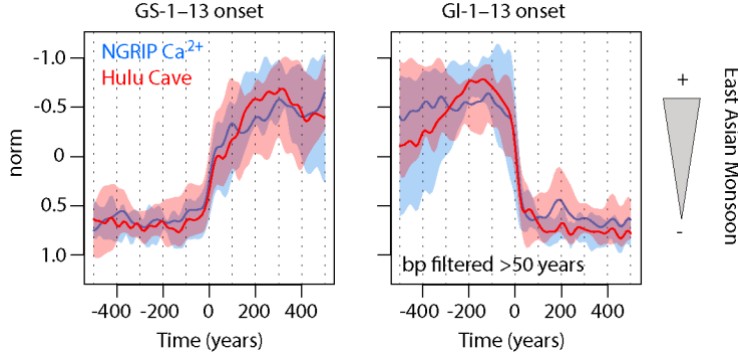

**Figure 3.** Stack of NGRIP Ca$^{2+}$ and Hulu Cave $\delta^{18}$O records using a technique in which 13 individual events are centered at the midpoint
of their abrupt transition, i.e. either DO warming (onset of GIs) or DO cooling (onset of GSs) (note the reverse scale). The events were
normalized and averaged to highlight the shared climatic signal at multidecadal and centennial timescales (>50 year low-pass filtered) and
compare the duration of the abrupt DO transitions between Greenland ice cores and Hulu Cave speleothems. Shading reflects the variability
across the events used for stacking. The midpoints of the abrupt transitions were identified using a Bayesian change-point analysis method
(Erdman and Emerson, 2007).






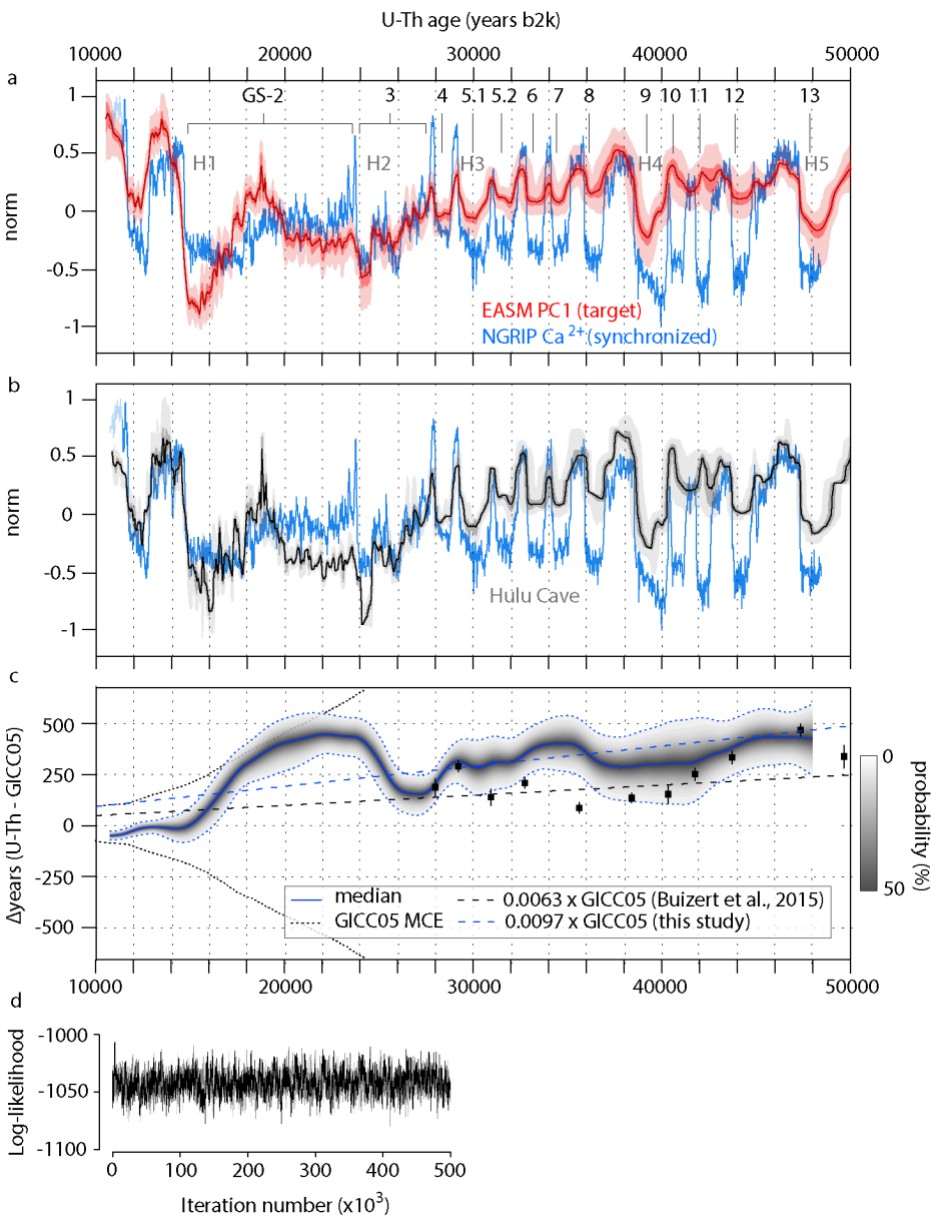

**Figure 4.** MCMC synchronization of GICC05 to the U-Th timescale during the last glacial period and resulting timescale transfer function. **a.** Synchronized Greenland $Ca^{2+}$ data on the U-Th timescale using the posterior median estimate of the MCMC synchronization. The synchronization was derived from stratigraphic alignment of the Greenland $Ca^{2+}$ data to the East Asian summer monsoon (EASM) PC1 (see Section 2.1 for details). Shadings reflect the empirical 68 and 95% confidence intervals from the 10,000 member ensemble. Greenland Stadials (GS) and timing of Heinrich Events (H) are indicated at the top. **b.** Comparison between the synchronized ice-core data and the Hulu Cave $\delta^{18}O$ record with its associated age uncertainty (grey shading: dark, 68%; dark 95%). All proxy records are shown in normalized





units. **c.** Posterior median (blue line) and pointwise 95% credible intervals (shading and blue dashed lines) of the difference $\Delta T$ between the GICC05 and U-Th timescales. The black squares are the Hulu–NGRIP age offsets presented in Buizert et al. (2015). The linear fit through

these data and that estimated from our $\Delta T$ values, are also shown. Note that the linear models are forced to intersect the origin. **d.** Progress of the MCMC algorithm showing 3 parallel chains (black, dark grey, grey) of the log-likelihood of the objective function.



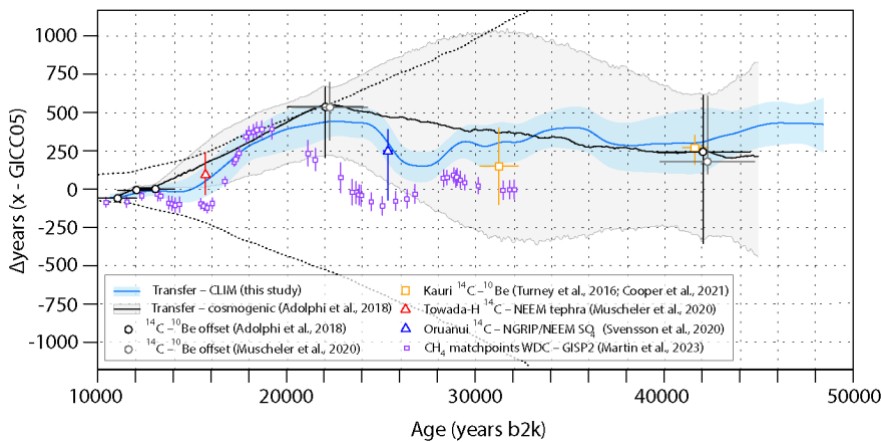

**Figure 5.** Posterior timescale transfer function based on the MCMC synchronization of the GICC05 to the U-Th timescale. Positive values indicate that the U-Th timescale is older than GICC05. The transfer function is presented with its median (thick lines) and pointwise 95% credible intervals (shading). The results are compared to the transfer function presented in Adolphi et al. (2018), which is based on a compilation of U-Th-dated $^{14}C$ records, including the low-resolution and less precisely dated Hulu Cave data (Southon et al., 2012). The markers with error bars ($\pm 2\sigma$) show discrete match points inferred from comparison of ice-core $^{10}Be$ records and absolutely dated $^{14}C$ data, $^{14}C$-dated volcanic eruptions identified in Greenland ice cores (Muscheler et al., 2020; Svensson et al., 2020), and methane match points between WAIS Divide and GISP2 ice cores (Martin et al., 2023). The dashed lines highlight the maximum counting uncertainty of GICC05.



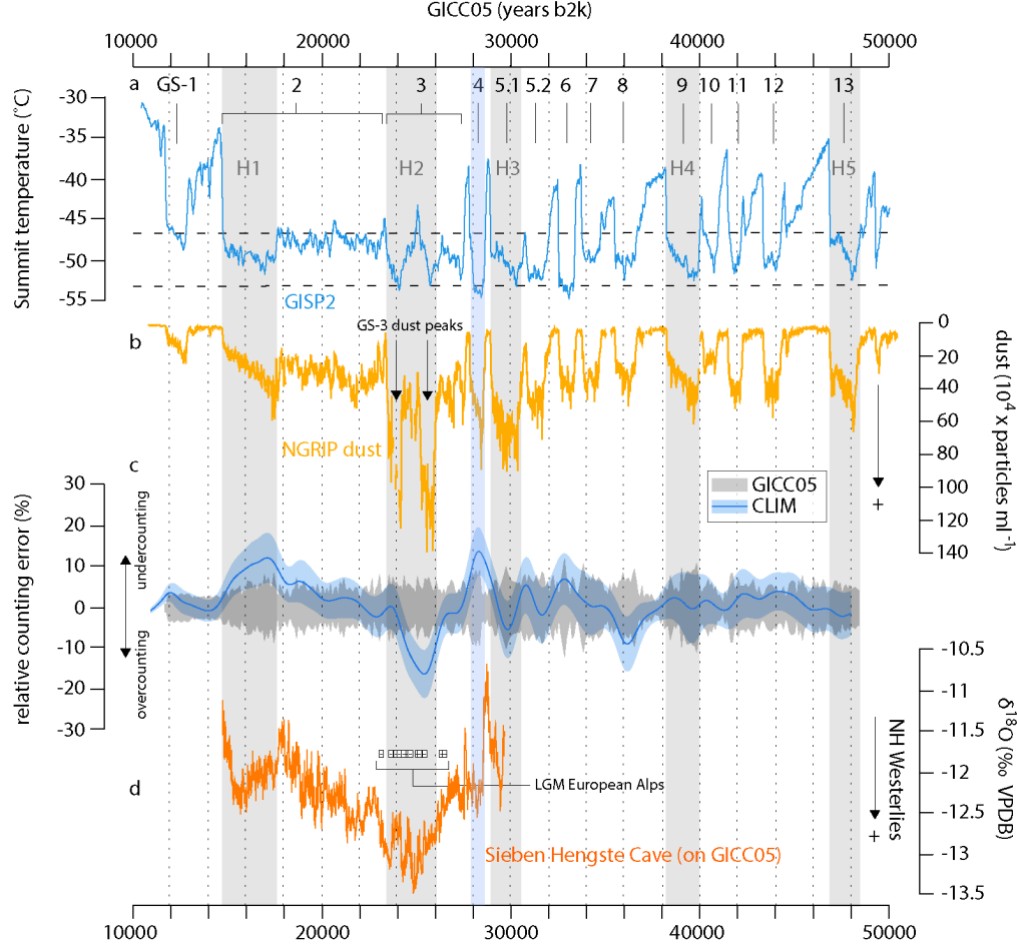

**Figure 6.** Inferred estimates of the relative annual-layer counting error for the GICC05 chronology based on MCMC synchronization to the U-Th timescale. **a.** GISP2 temperature reconstruction NGRIP annual layer thickness (Martin et al., 2023) presented with partitioning of Greenland Stadials (GS) and timing of Heinrich Events (H; grey vertical bars). Underlying dashed levels show the ±2σ temperature range of stadials GS-1 to GS-13. Interstadial-stadial transitions were identified using a Bayesian change point procedure (Erdman and Emerson, 2007). The blue vertical bar denotes GS-4, i.e. the coldest stadial (3.3˚C colder than the average). **b.** NGRIP insoluble dust concentration record (Ruth et al., 2003) (note the reverse scale). **c.** Comparison between the maximum relative counting error of the GICC05 (grey shading) and the differential dating uncertainties inferred from the CLIM synchronization, respectively, presented with their posterior median (thick lines) and pointwise 68% credible intervals (shading). Positive (negative) values indicate an undercount (overcount) of ice layers in Greenland ice cores. **d.** U-Th dated climate records of the Last Glacial Maximum (LGM) in the European Alps after synchronization to the GICC05 timescale by applying the transfer function presented in this study. U-Th ages of cryogenic cave carbonates (Spötl et al., 2021) with their ±2σ uncertainty (white squares) indicating the timing of the maximum mountain glacier extent over the European Alps, and δ18O values of precipitation from the Sieben Hengste stalagmite record (Luetscher et al., 2015) (orange). The Sieben Hengste record reveals a maximum strengthening and southerly displacement of the westerly winds during GS-3. All records are presented on the GICC05 timescale.