# Peer review of "Continuous synchronization of the Greenland ice-core and U-Th timescales using probabilistic inversion"

_Climate of the Past, 2023_

## Author Comment (AC1)

We thank Reviewer1 for the constructive comments and detailed scrutiny of our manuscript. We agree that there are points that can be further clarified and we are happy to address the main and specific points that the Reviewer has brought up.

**General comments**
[…] This study goes one step further, as it has the underlying assumption that all climate records are indeed the same (equation in line 228). Spelled out, this means that NGRIP Ca2+ = Speleothem d18O, which is obviously not true as they are very different physical quantities, controlled by different processes. Admittedly, some of the controlling processes may be shared, but the assumption of the applied method is much stronger: It implies the existence of a linear function that relates NGRIP Ca2+ and speleothem d18O. […] the basic assumption underlying the approach presented here (i.e., NGRIP Ca2+ = speleothem d18O) is incorrect and should hence not be used.

We appreciate the Reviewer raising this point, as it allows us to further clarify details of the method. The notation in the equation indicates an approximation ($\approx$) rather than an equality. It is also important to note that this assumption is performed after scaling both data sets to have a strict range of [-1,1], which means $\forall\, t \in T : -1 \leq u(t) \leq 1 \land -1 \leq g(\tau(t)) \leq 1$, where T is the window range of the target (this will be clearly explained in the revision).

We agree that a nonlinear alignment between the records is difficult to capture with a single linear function. However, as described in the manuscript, we employ a piecewise linear function to model the alignment between the NGRIP Ca2+ and speleothem $\delta^{18}O$ records. A piecewise linear function is fundamentally different than a simple linear function, as it works by dividing the domain into K separate intervals and fitting a different linear segment within each interval. The linear segments are connected at breakpoints, where the slope can change. This enables the piecewise function to locally approximate nonlinear patterns by using linear segments that capture the essence of more complex functional forms in each region of the domain.

In fact, using this segmented piecewise linear technique provides several key advantages compared to a simple global linear function:

- It allows flexible modelling of nonlinear shapes by adapting the slope in each interval.
- Computationally efficient optimization algorithms can be applied by leveraging the linear segments. A purely nonlinear function would be more challenging to optimize.
- Accuracy can be systematically improved by adding more segments. The nonlinearity is approximated to any desired level.
- The approach balances accuracy with efficiency. More complex nonlinear functions could overfit given uncertainties in the data.

In summary, the piecewise linear methodology enables tractably approximating the nonlinear alignment relationship while facilitating optimization. We understand how these nuanced distinctions may have been unclear in the original manuscript, and we welcome this opportunity to explain our innovative approach more fully in the revised version of the manuscript.

If the authors nonetheless want to follow this approach, they need to i) clearly state that their model assumptions are not fulfilled…

Please see our detailed reply above. We will discuss our approach in more detail in the revised version of the manuscript.

…and ii) discuss these drawbacks and provide additional tests to demonstrate the robustness of the results.

This is a good idea and we are happy to provide two new Δt transfer functions based on NGRIP and GRIP $\delta^{18}O$ records, respectively. The new results are internally coherent and support the findings obtained using NGRIP Ca2+, which ultimately lend strength to our conclusions. These results will be incorporated and discussed in the new version of the manuscript.

1. What determines the inferred timescale shift during the LGM, when there is little co-variability between NGRIP Ca and EASM PC1 (see figure 4)?

Between 18-24ka there is little align-able structure in the timeseries and in fact the model sticks to the information obtained outside this interval, i.e. effectively the Δt does not move much, and the alignment uncertainty grows accordingly. This is to be expected and in line with the design of our alignment model.

2. Which timescale offset is inferred when only the period between 15 – 22 kaBP (or other subsections) is synchronized (and both records are standardized only for this period)?

Unfortunately, the method is not built to align very short timeseries with little structure and low signal-to-noise ratios. We hope that this issue is resolved by providing additional synchronization tests and targets using NGRIP and GRIP $\delta^{18}O$, which demonstrate that the Δt estimates are overall robust during this period. In addition, the Δt during this critical interval is corroborated by independent estimates published by Dong et al., 2022 (see Reviewer2's suggestions/comments). This will be clarified in the new version of the manuscript.

3. How would the results differ if NGRIP d18O was used instead of Ca2+?

Thanks for this suggestion. This is a sensible request and more in line with the premise of our manuscript, i.e. we show the physical relationship between Greenland air temperature and precipitation in the EASM region (see Fig. 1). The results are qualitatively consistent with those based on NGRIP Ca2+, which, again, demonstrates that our method is overall robust (please see our replies above).

4. How would the results (and uncertainties) differ if the uncertainty sigma_ui in the model was increased sufficiently to fulfil the model assumption (NGRIP Ca2+ = speleothem d18O within error).

Thanks for bringing this up. It should be noted that the input and target are scaled between -1 and 1, so by using a 0.1 stdev we are effectively covering 30% of the observable window, and on top of that we are using a heavy tailed distribution (t-distro) which means that we are assuming an uncertainty that fulfils the NGRIP Ca2+ = speleothem $\delta^{18}O$ assumption. We should also mention that we are employing overly conservative estimates for sigma using a multiplying factor of 2 (this will be clarified in the revised version of the manuscript).

Further, the results need to be evaluated more critically with respect to previous studies:

1. Please include the timescale differences by Corrick et al. into figures 4 & 5.

We will include Corrick's Δt estimates in the new version of the manuscript.

2. It appears that most other studies (Buizert et al. / Corrick et al. / Martin et al) found systematically smaller timescale differences then the results presented here. Why?

We can certainly mention this in the manuscript. The small differences may stem from the fact that previous studies used only Hulu Cave $\delta^{18}O$ data, whereas here we use a more comprehensive approach that incorporates several EASM spelothem records.

**Specific comments**
L205 (eq. 1): Maybe I got this wrong but looking at this equations and trying to put in [units]: m must be [years/m]; so t must be [m] not time; so tau is defined on depth? If so, please use a different symbol as t is time later on.

We appreciate the reviewer raising this insightful question about the units in equation 1. It allows us to clarify that in our case, m is dimensionless, representing an expansion/compaction parameter in units of years/years. Meanwhile, $\tau_0$, $\delta$, and $c_i$ have units of years. This will be clarified in the revised version of the manuscript.

L180: "in response to changes in accumulation" since you're not modelling accumulation, maybe better "in response to miscounting"? L213-214: "…distinct depositional environments…" But you only model the ice core alignement and their accumulation rates are certainly autocorrelated? It is ok to do it like that, but I am not sure I agree with the explanation. L214-216: Isn't it that: You are not modelling the timescale (or ice accumulation) but only minor modifications of it (counting errors), which do not need to be autorcorrelated.

This is correct. Thank you for point this out. Instead of using the term "accumulation rates" we will discuss the model results in terms of compaction/expansion of the original timescale.

L228: "u(ti) = g(tau(ti))" See major comments. This is obviously not true and should be discussed.

As we mentioned in our response to the previous question regarding equation 1, we agree it is crucial to use clear and consistent notation to avoid confusion. The reviewer is correct that "u(ti) = g(tau(ti))" is imprecise shorthand and could be misinterpreted. Nevertheless, we use the notation $u(ti) \approx g(\tau(ti))$, which means $\exists \tau(z) \mid u(zi) \approx g(\tau(zi))$. Note that we use $\approx$ and not =.

L244 (Eq. 3): This was defined for comparing 14C-dates to a 14C-calibration. I.e., similar physical quantities. Because your sigma_ui is too small to fulfil the model (u=g) the vast majority of the data is essentially treated as outliers in the gamma-distribution. See major comments.

It is important to note that u(ti) and $g(\tau(ti))$ represent the NGRIP Ca2+ and aligned speleothem δ18O records, respectively, after rescaling the data to the interval [-1,1]. This

rescaling means that the uncertainty $\sigma\_ui$ used in the t-distribution becomes a conservative estimate around the rescaled record $u(z_i)$. However, the use of the t-distribution has proven robust against outliers. Therefore, any data points that become outliers due to the rescaling assumptions do not significantly affect the resulting inferences of the alignment function $\tau(z)$. The t-distribution's heavy tails downweigh the influence of extreme values. In summary, rescaling the records to [-1,1] provides a simple standardized domain for comparing the data, while the t-distribution likelihood protects against artifacts from this transformation when inferring the optimal $\tau(z)$. This will be clarified in the new version of the manuscript.

L331-332: The agreement between the records is not very convincing. Please discuss critically. What is the correlation coefficient? What is the error of the model (u=g) after alignment?

We appreciate the reviewer's feedback on discussing agreement between the aligned records. However, we deem applying traditional metrics like a correlation coefficient unnecessary in our Bayesian alignment framework. The optimization process inherently identifies the maximum likelihood alignment given the uncertainties around each data point $u(t_i)$ and $g(\tau(t_i))$.

Specifically, in each step of inferring the posterior distribution for the piecewise linear function $\tau(t)$, the likelihood is calculated based on the t-distribution residuals between $u(t_i)$ and $g(\tau(t_i))$. The Bayesian approach thus quantitatively determines the optimal nonlinear alignment that maximizes the joint likelihood. Therefore, rather than introducing additional metrics, we believe the optimal uncertainties around $\tau(z)$ themselves demonstrate the credible alignment between the records.

**Other specific comments**

All the other minor comments and suggestions raised by the Reviewer will be respected in our revised manuscript.

---

## Author Comment (AC2)

We thank Reviewer2 for the supportive review and the constructive comments. We are happy to accept R2's suggestions and meet all their requests.

**General comments**
[…] I generally agree to the finding of this and previous studies that there is some quite strong bias in the GICC05 layer counting for the 15-28 ka section that was fairly unconstrained at the time. In some sections, the bias appears larger than the stated MCE, and quite likely, the bias goes in both directions for different periods ending up at a close-to-correct absolute age for much of the 30-40 ka section. Still, I would think there is also the possibility that the U-Th stalagmite ages may sometimes have their accuracy issues although they are often published with very small error bars. Alone the observed scatter among different stalagmites covering the same events points in this direction. I think we have an example of this for the applied stalagmite records at around GI-10, where they 'exhibit some temporal inconsistencies' (Figure 4). Therefore, I would be careful to assume that all of the observed disagreement in absolute ages between the ice core and U-Th chronologies can be attributed issues related to the ice-core time scale(s). In any case, a long-term absolute error of about 1% is certainly much smaller than we thought it possible some 15-20 years ago, when GICC05 was put together.

This is a fair point and we agree with the Reviewer. We welcome this opportunity to tone down our claims and stress that the U-Th timescale (although absolute) may be problematic in certain intervals. We will discuss this potential issue more prominently in the revised version of the manuscript.

**The following recent papers may be relevant to mention or discuss in the manuscript:**

Dong et al., 2022, is concerned with GS-3 and introduces some accurately dated Asian stalagmites that allow for a detailed comparison of ice core and U-Th ages across that interval. The paper is supportive of the ice-core Ca/dust – Asian monsoon relationship for significant and abrupt climate events and it identifies biases of the ice-core chronologies in the same direction as the present manuscript although with somewhat smaller amplitudes.

Sinnl et al., 2023, identifies new 10Be bipolar links between G and A in the older part of the difficult GS-2 interval. The study is thus relevant for comparison in a similar way to that of Martin et al., 2023.

Many thanks for the suggestions. We will certainly discuss these studies in the revised version of the manuscript.

**Specific comments:**

Lines 331-341: To test the robustness of the suggested similarity of the Greenland and East Asian records across GS-2 it may be an idea to apply a different Greenland record for the inversion algorithm.

We appreciate the Reviewer raising this point as it was brought up by R1 as well. This is a good suggestion and we are happy to provide two new $\Delta t$ transfer functions based on NGRIP and GRIP $\delta^{18}O$ records (as recommended by R1), respectively. As discussed in our replies to R1, the new transfer functions are consistent and overall support the results based on NGRIP Ca2+. These new findings will be presented in the new version of the manuscript.

Figure 6: In the attached figure, I compare the Sieben Hengste Cave (SHC) isotope record to the Ca and dust profiles of NGRIP and NEEM (all ice core records are on log scales). The SHC record is shown on its original time scale without application of the transfer function. Shown on those time scales, there appears to be a good correspondence between the ice core records and the SHC isotopes for the 22-28 ka period. In particular, the sharp transition associated with the onset of the younger of the Greenland dust spikes close to 24 ka and the adjacent structures seem to be well aligned between all records. Therefore, assuming there is a one-to-one relationship between ice-core dust/Ca and European stalagmite d18O, it appears that the transfer function makes things worse for this interval. If there are common events between the two records at around 18 ka, the transfer function may do a better job here?

We appreciate the Reviewer taking the time to compare SHC $\delta^{18}$O to Greenland ice core data. Estimating the offset between SHC $\delta^{18}$O and GICC05 using our methodology is an interesting suggestion but somewhat beyond the scope of this study. We are concerned that the SHC $\delta^{18}$O reflects a compound signal of changes in atmospheric circulation and moisture advection pathways that is not as physically well understood as for the EASM speleothem records (e.g. Luetscher et al., 2015). We should also point out that any mismatch between SHC $\delta^{18}$O and Greenland records may be an expression of dating uncertainties associated with assumptions about growth rates, interpolation models, purity of U-Th samples, etc. The limitations of over-emphasizing one record for synchronization purposes has been a key matter of discussion during a previous iteration of the current manuscript, and the reason why we now "stack" several $\delta^{18}$O records rather than relying on one record (e.g. Hulu Cave). For these reasons we prefer to use the SHC $\delta^{18}$O data only for a qualitative comparison to the long-term biases in RCEs presented in Fig. 6.

**Other specific comments**

All the other minor comments and suggestions raised by the Reviewer will be respected in our revised manuscript.

---

## Author Response (AR1)

We thank Reviewer1 for the constructive comments and detailed scrutiny of our manuscript. We agree that there are points that can be further clarified and we are happy to address the main and specific points that the Reviewer has brought up.

**General comments:**

[…] This study goes one step further, as it has the underlying assumption that all climate records are indeed the same (equation in line 228). Spelled out, this means that NGRIP Ca2+ = Speleothem d18O, which is obviously not true as they are very different physical quantities, controlled by different processes. Admittedly, some of the controlling processes may be shared, but the assumption of the applied method is much stronger: It implies the existence of a linear function that relates NGRIP Ca2+ and speleothem d18O. […] the basic assumption underlying the approach presented here (i.e., NGRIP Ca2+ = speleothem d18O) is incorrect and should hence not be used.

We appreciate the Reviewer raising this point, as it allows us to further clarify details of the method. The notation in the equation indicates an approximation ($\approx$) rather than an equality. It is also important to note that this assumption is performed after scaling both data sets to have a strict range of [-1,1], which means $\forall\, t \in T : -1 \leq u(t) \leq 1 \wedge -1 \leq g(\tau(t)) \leq 1$, where $T$ is the window range of the target (this will be clearly explained in the revision).

We agree that a nonlinear alignment between the records is difficult to capture with a single linear function. However, as described in the manuscript, we employ a piecewise linear function to model the alignment between the NGRIP Ca2+ and speleothem $\delta^{18}O$ records. A piecewise linear function is fundamentally different than a simple linear function, as it works by dividing the domain into K separate intervals and fitting a different linear segment within each interval. The linear segments are connected at breakpoints, where the slope can change. This enables the piecewise function to locally approximate nonlinear patterns by using linear segments that capture the essence of more complex functional forms in each region of the domain.

In fact, using this segmented piecewise linear technique provides several key advantages compared to a simple global linear function:

- It allows flexible modelling of nonlinear shapes by adapting the slope in each interval.
- Computationally efficient optimization algorithms can be applied by leveraging the linear segments. A purely nonlinear function would be more challenging to optimize.
- Accuracy can be systematically improved by adding more segments. The nonlinearity is approximated to any desired level.
- The approach balances accuracy with efficiency. More complex nonlinear functions could overfit given uncertainties in the data.

In summary, the piecewise linear methodology enables tractably approximating the nonlinear alignment relationship while facilitating optimization. We understand how these nuanced distinctions may have been unclear in the original manuscript, and we welcome this opportunity to explain our innovative approach more fully in the revised version of the manuscript (new lines 217-225, and 244-246).

If the authors nonetheless want to follow this approach, they need to i) clearly state that their model assumptions are not fulfilled…

Please see our detailed reply above. We discussed our approach in more detail in the revised version of the manuscript.

…and ii) discuss these drawbacks and provide additional tests to demonstrate the robustness of the results.

This is a good idea and we are happy to provide two new Δt transfer functions based on NGRIP and GRIP $\delta^{18}$O records, respectively. The new results are internally coherent and support the findings obtained using NGRIP Ca2+, which ultimately lend strength to our conclusions. These results are incorporated and discussed in the new version of the manuscript (new Figures 2, 5, 6, 7, 8 and new Section 2.1 and 3).

**1. What determines the inferred timescale shift during the LGM, when there is little co-variability between NGRIP Ca and EASM PC1 (see figure 4)?**

Between 18-24ka there is little align-able structure in the timeseries and in fact the model sticks to the information obtained outside this interval, i.e. effectively the Δt does not move much, and the alignment uncertainty grows accordingly. This is to be expected and in line with the design of our alignment model.

**2. Which timescale offset is inferred when only the period between 15 – 22 kaBP (or other subsections) is synchronized (and both records are standardized only for this period)?**

Unfortunately, the method is not built to align very short timeseries with little structure and low signal-to-noise ratios. We hope that this issue is resolved by providing additional synchronization tests and targets using NGRIP and GRIP $\delta^{18}$O, which demonstrate that the Δt estimates are overall robust during this period. In addition, the Δt during this critical interval is corroborated by independent estimates published by Sinnl et al. (2023) and Dong et al. (2022) (see also R2's suggestions/comments). This is highlighted in the new version of the manuscript (new lines 349-351, and 393-395).

**3. How would the results differ if NGRIP d18O was used instead of Ca2+?**

Thanks for this suggestion. This is a sensible request and more in line with the premise of our manuscript, i.e. we show the physical relationship between Greenland air temperature and precipitation in the EASM region (see Fig. 1). The results are qualitatively consistent with those based on NGRIP Ca2+, which, again, demonstrates that our method is overall robust (please see our replies above). The new results are shown in Figures 5, 6, 7, 8 and discussed in the new sections 2.1 and 3.

**4. How would the results (and uncertainties) differ if the uncertainty sigma_ui in the model was increased sufficiently to fulfil the model assumption (NGRIP Ca2+ = speleothem d18O within error).**

Thanks for bringing this up. It should be noted that the input and target are scaled between -1 and 1, so by using a 0.1 stdev we are effectively covering 30% of the observable window, and on top of that we are using a heavy tailed distribution (t-distro) which means that we are assuming an uncertainty that fulfils the NGRIP Ca2+ = speleothem $\delta^{18}$O assumption. We should also mention that we are employing overly conservative estimates for sigma using a multiplying factor of 2 (this is now clarified in the revised version of the manuscript, new line 253-257).

Further, the results need to be evaluated more critically with respect to previous studies:

1. Please include the timescale differences by Corrick et al. into figures 4 & 5.

We have included Corrick's Δt estimates in the new Figure 7.

2. It appears that most other studies (Buizert et al. / Corrick et al. / Martin et al) found systematically smaller timescale differences then the results presented here. Why?

The difference is marginal and Corrick's estimates are simply too uncertain to ascertain whether the offset is meaningful/systematic (see new Figure 7). As to the other studies, the small differences may stem from the fact that previous work used only Hulu Cave $\delta^{18}O$ data, whereas here we use a more comprehensive approach that incorporates several EASM spelothem records. In addition, the new synchronizations based on NGRIP and GRIP d18O suggest that the bias may be smaller than previously estimated, i.e. possibly 0.75% (new lines 338-341).

**Specific comments:**

L205 (eq. 1): Maybe I got this wrong but looking at this equations and trying to put in [units]: m must be [years/m]; so t must be [m] not time; so tau is defined on depth? If so, please use a different symbol as t is time later on.

We appreciate the reviewer raising this insightful question about the units in equation 1. It allows us to clarify that in our case, m is dimensionless, representing an expansion/compaction parameter in units of years/years. Meanwhile, $\tau_0$, $\delta$, and ci have units of years. This is now clarified in the revised version of the manuscript (new lines 187, 208-209).

L180: "in response to changes in accumulation" since you're not modelling accumulation, maybe better "in response to miscounting"? L213-214: "…distinct depositional environments…" But you only model the ice core alignment and their accumulation rates are certainly autocorrelated? It is ok to do it like that, but I am not sure I agree with the explanation. L214-216: Isn't it that: You are not modelling the timescale (or ice accumulation) but only minor modifications of it (counting errors), which do not need to be autorcorrelated.

This is correct. Thank you for point this out. Instead of using the term "accumulation rates" we now discuss the model results in terms of compaction/expansion of the original timescale (new lines 189, 260).

L228: "u(ti) = g(tau(ti))" See major comments. This is obviously not true and should be discussed.

As we mentioned in our response to the previous question regarding equation 1, we agree it is crucial to use clear and consistent notation to avoid confusion. The reviewer is correct that "u(ti) = g(tau(ti))" is imprecise shorthand and could be misinterpreted. Nevertheless, we use the notation u(ti) ≈ g($\tau$(ti)), which means ∃ $\tau$(z) | u(zi) ≈ g($\tau$(zi)). Note that we use ≈ and not =.

L244 (Eq. 3): This was defined for comparing 14C-dates to a 14C-calibration. I.e., similar physical quantities. Because your sigma_ui is too small to fulfil the model (u=g) the vast majority of the data is essentially treated as outliers in the gamma-distribution. See major comments.

It is important to note that $u(t_i)$ and $g(\tau(t_i))$ represent the NGRIP Ca2+ and aligned speleothem $\delta18O$ records, respectively, after rescaling the data to the interval [-1,1]. This rescaling means that the uncertainty $\sigma\_{ui}$ used in the t-distribution becomes a conservative estimate around the rescaled record $u(z_i)$. However, the use of the t-distribution has proven robust against outliers. Therefore, any data points that become outliers due to the rescaling assumptions do not significantly affect the resulting inferences of the alignment function $\tau(z)$. The t-distribution's heavy tails downweigh the influence of extreme values. In summary, rescaling the records to [-1,1] provides a simple standardized domain for comparing the data, while the t-distribution likelihood protects against artifacts from this transformation when inferring the optimal $\tau(z)$. This is now clarified in the new version of the manuscript (new lines 253-257).

L331-332: The agreement between the records is not very convincing. Please discuss critically. What is the correlation coefficient? What is the error of the model (u=g) after alignment?

We appreciate the reviewer's feedback on discussing agreement between the aligned records. However, we deem applying traditional metrics like a correlation coefficient unnecessary in our Bayesian alignment framework. The optimization process inherently identifies the maximum likelihood alignment given the uncertainties around each data point $u(t_i)$ and $g(\tau(t_i))$.

Specifically, in each step of inferring the posterior distribution for the piecewise linear function $\tau(t)$, the likelihood is calculated based on the t-distribution residuals between $u(t_i)$ and $g(\tau(t_i))$. The Bayesian approach thus quantitatively determines the optimal nonlinear alignment that maximizes the joint likelihood. Therefore, rather than introducing additional metrics, we believe the optimal uncertainties around $\tau(z)$ themselves demonstrate the credible alignment between the records.

**Other specific comments:**

L13-14: "which are currently not detectable…" Why wouldn't they be?
This has been re-worded.

L19-20: "a bias attributable…" The paper provides a reasonable discussion around this, but it is not conclusive. Please add "possibly" or similar.
This has been revised accordingly.

L37-38: "much smaller uncertainty in the absolute ages". During the glacial.
This has been revised accordingly.

L95-96: Please also mention the advantage, that this is a relatively low-level assumption, that only requires synchroneity and not a linear relationship as assumed by the model applied here. Further, the discrete tie-points have typically a high signal to noise ratio, while the method applied here, employs also low signal to noise variations for matching. Please be critical with the assumptions of your method.
We rephrased this paragraph and now acknowledge the high signal to noise ratio of abrupt proxy transitions (new lines 96-99).

L103-111: In principle, I agree with the problems of the alignment technique, but I am not sure what this has to do with (which?) autocorrelation. 10Be is autocorrelated over time - so are most climate and forcing records. The autocorrelation argument would also be true for pure 14C-wiggle match-dating. In my opinion, the crux is the window-length: If there is one large

peak within a window, it will dominate the obtained pdf as long as it is in the window. Hence, we used only non-overlappting windows in Adolphi & Muscheler 2016. But that obviously affects the resolution we can obtain, as we need a certain window length to have a signal.
We deem this sufficiently clear but we are happy to take on specific suggestions from the Reviewer as to how we can edit the text.

L117-119: This is not an issue of the alignment technique, but of the lack of convincing tie-points. In Adolphi et al. 2018 we only chose one tie-point around 21 kaBP which we called "tentative" but which forms the basis of much of what is discussed here. The lack of tie-points (or co-variability) is similar in this study. Looking at figure 4 there seems little agreement between the records. See major comments.
We deleted this comment accordingly.

L143-145: There are many processes that contribute to Ca deposition in Greenland. Please discuss in more detail.
We now acknowledge these additional processes this (new line 144).

L167-169: Maybe point out the advantages too: The assumption that the timing of a major climate transition synchronous is much more conservative than assuming a linear relationship between the proxies on all timescales which is clearly proven wrong during the LGM.
We edited the text accordingly (new line 173-174).

L180: "in response to changes in accumulation" since you're not modelling accumulation, maybe better "in response to miscounting"?
Thanks. This has been changed throughout the manuscript.

L181-182: "simulated ice core depositional history". See above, you're not modelling deposition but only the timescale.
This has been changed accordingly (new lines 189).

L205 (eq. 1): Maybe I got this wrong but looking at this equations and trying to put in [units]: m must be [years/m]; so t must be [m] not time; so tau is defined on depth? If so, please use a different symbol as t is time later on.
This has been addressed above (see specific comments).

L213-214: "…distinct depositional environments…" But you only model the ice core alignement and their accumulation rates are certainly autorcorrelated? It is ok to do it like that, but I am not sure I agree with the explanation.
Thanks. We removed this line to avoid confusion.

L214-216: Isn't it that: You are not modelling the timescale (or ice accumulation) but only minor modifications of it (counting errors), which do not need to be autorcorrelated.
That is correct. Thank you. We changed the wording throughout the manuscript to stress this out.

L228: "u(ti) = g(tau(ti))" See major comments. This is obviously not true and should be discussed.
This has been addressed. Please see our response above.

L244 (Eq. 3): This was defined for comparing 14C-dates to a 14C-calibration. I.e., similar physical quantities. Because your sigma_ui is too small to fulfil the model (u=g) the vast majority of the data is essentially treated as outliers in the gamma-distribution. See major comments.
This has been addressed. Please see our response above.

L257: "integrates" better "reflects" as this is the derivative of the MCE which may cause confusion.
Thanks. This has been changed.

L258-259: but that shift is absolute? Why is the RCE a good measure here?
Apologies for the confusion. The initial shift is in fact absolute. We edited the text accordingly.

L268-269: "exceeds the range allowed by the MCE (as is generally the case for the Holocene)". This is not true. We also discuss, that the RCE is only exceeded very briefly. The exceeded MCE is inhereted from this early mistake. See figure 12 in Adolphi and Muscheler 2016.
Thank you. This statement has been deleted.

L318: 0.97 is 50% more than 0.63! Is that "comparable"?
The new results based on ice-core d18O are more in line with Buizert's findings (now discussed in lines 338-339).

L321: Please compare the Delta T to Muscheler et al. 2008
This is now discussed in the main text (lines 345-346).

L322: "younger" within error this is consistent?
Apologies, we don't understand this comment.

L331-332: See major comments. The agreement between the records is not very convincing. Please discuss critically. What is the correlation coefficient? What is the error of the model (u=g) after alignment?
This has been discussed in our reply above.

L334-335: "the error is large". It appears that the error is actually smaller than during MIS-3?
This line has been edited accordingly.

L345: There seems to be quite some disagreement with Martin et al. 2023. Please discuss.
To avoid confusion due to comparing multiple independent timescales, we decided to remove these data entirely and only focus on the Δt between the U-Th and GICC05 timescales.

L346: Please include the re-assessment of the LGM tie-point by Sinnl et al., (2023) into the figures
This information has been added in the main text and on new Figure 7.

L361: Please also cite Sinnl et a. (2023)
This has been addressed.

Figure 4: Please include Corrick et al. 2023 tie-points
This has been addressed (see new Fig. 7).

####################################################################################
####################################################################################

We thank Reviewer2 for the supportive review and the constructive comments. We are happy to accept R2's suggestions and meet all their requests.

agree that there are points that can be further clarified and we are happy to address the main and specific points that the Reviewer has brought up.

**General comments:**

[…] I generally agree to the finding of this and previous studies that there is some quite strong bias in the GICC05 layer counting for the 15-28 ka section that was fairly unconstrained at the time. In some sections, the bias appears larger than the stated MCE, and quite likely, the bias goes in both directions for different periods ending up at a close-to-correct absolute age for much of the 30-40 ka section. Still, I would think there is also the possibility that the U-Th stalagmite ages may sometimes have their accuracy issues although they are often published with very small error bars. Alone the observed scatter among different stalagmites covering the same events points in this direction. I think we have an example of this for the applied stalagmite records at around GI-10, where they 'exhibit some temporal inconsistencies' (Figure 4). Therefore, I would be careful to assume that all of the observed disagreement in absolute ages between the ice core and U-Th chronologies can be attributed issues related to the ice-core time scale(s). In any case, a long-term absolute error of about 1% is certainly much smaller than we thought it possible some 15-20 years ago, when GICC05 was put together.

This is a fair point and we agree with the Reviewer. We welcome this opportunity to tone down our claims and stress that the U-Th timescale (although absolute) may be problematic in certain intervals. We discuss this potential issue more openly in the revised version of the manuscript (new lines 383-386).

**The following recent papers may be relevant to mention or discuss in the manuscript:**

Dong et al., 2022, is concerned with GS-3 and introduces some accurately dated Asian stalagmites that allow for a detailed comparison of ice core and U-Th ages across that interval. The paper is supportive of the ice-core Ca/dust – Asian monsoon relationship for significant and abrupt climate events and it identifies biases of the ice-core chronologies in the same direction as the present manuscript although with somewhat smaller amplitudes.

Sinnl et al., 2023, identifies new 10Be bipolar links between G and A in the older part of the difficult GS-2 interval. The study is thus relevant for comparison in a similar way to that of Martin et al., 2023.

Many thanks for the suggestions. We now mention these studies in the revised version of the manuscript (new section 3.1 and 3.2) and present the data from Sinnl et al. in new Fig. 7. The offset estimate from Dong et al. (i.e. +320 years) is in good agreement with our new results. Our estimates integrated over the same 5-kyr period suggest a shift of +335 years for CLIM1, +255 years for CLIM2, and +240 years for CLIM3. The difference is mainly due to averaging the structure of our $\Delta t$ transfer functions over five millennia. As for Sinnl et al., they estimated an offset of +375 years around 22 kyr b2k, which is remarkably similar to our $\Delta t$ estimate of +390 years (mean of the three synchronizations; please see new Section 3.1).

**Specific comments:**

Lines 331-341: To test the robustness of the suggested similarity of the Greenland and East Asian records across GS-2 it may be an idea to apply a different Greenland record for the inversion algorithm. The NGRIP dust record is available in 5cm resolution, but it has rather poor quality when it comes to details. NEEM has available high-resolution records available for both Ca and dust concentrations. It may be worth trying to match the dust record and maybe the Ca using a log scale as the dust concentration varies exponentially with Greenland water isotopes (see attached figure).

We appreciate the Reviewer raising this point as it was brought up by R1 as well. This is a good suggestion and we will provide two new $\Delta t$ transfer functions based on NGRIP and GRIP $\delta^{18}O$ records, respectively. As discussed in our replies to R1, the new transfer functions are consistent and overall support the results based on NGRIP Ca2+. These new findings are

presented in the new version of the manuscript (please see replies to R1's comments). Also please note that the Ca2+ data have been log transformed before synchronization (see Fig. 2a).

Figure 6: In the attached figure, I compare the Sieben Hengste Cave (SHC) isotope record to the Ca and dust profiles of NGRIP and NEEM (all ice core records are on log scales). The SHC record is shown on its original time scale without application of the transfer function. Shown on those time scales, there appears to be a good correspondence between the ice core records and the SHC isotopes for the 22-28 ka period. In particular, the sharp transition associated with the onset of the younger of the Greenland dust spikes close to 24 ka and the adjacent structures seem to be well aligned between all records. Therefore, assuming there is a one-to-one relationship between ice-core dust/Ca and European stalagmite d18O, it appears that the transfer function makes things worse for this interval. If there are common events between the two records at around 18 ka, the transfer function may do a better job here?

We think that the SHC $\delta^{18}O$ is still marginally older than GICC05, although this is difficult to quantify with the naked eye. In particular, the structure around 28-30kaBP lends support to a systematically older U-Th timescale than GICC05. Estimating the offset using our methodology is an interesting suggestion but somewhat beyond the remit of this study. We believe this approach would better fit the scope of a follow-up project. Specifically, we are concerned that the SHC $\delta^{18}O$ data reflects a compound signal of atmospheric circulation that is not as physically straightforward (or, by all means sufficiently well understood) as for the EASM speleothem record.

**Specific comments:**

Lines 242-248: This may be a good place also to discuss the Sinnl et al., 2023, bipolar 10Be match points. Please also elaborate a bit on the relevance of the Martin et al, 2023, study. It may not be evident for the reader why the G – A synchronization is relevant in a context that is otherwise entirely NH.
Thank you for pointing this out. As explained in our response to R1, we decided to remove the data from Martin et al., to avoid confusion and comparing multiple independent timescales. Rather we prefer to focus entirely on the differences between the U-Th and GICC05 timescales.

The results from Sinnl et al. have been incorporated in the main text (please see our previous resplies) and presented in Fig. 7.

Line 361: Clearly, there is god agreement between the results of the present study and that of Martin et al., 2023, at around 18 ka in Figure 5, but for younger and particularly for older ages, there are large discrepancies, so what are the implications of this? Again, it may not be evident to the reader how Antarctica fits into the otherwise NH picture. The Dong et al., 2022, study could be relevant for this discussion.
Thank you. Please see our comment above. The results from Dong et al. have been incorporated in the main text.

Figure 3: A convincing comparison (although not surprising) but something must be wrong with figure titles or the caption. Should be right-hand figure be showing GS onsets? Not sure which reversed scale is referred to in the caption.
Thanks for pointing this out. We acknowledge that the time axis may be confusing. The figure has now been edited so that time plots right-to-left.

Figure 4 caption: Which blue line is referred to in caption of Figure 3c? In Figure 3d, I can also not distinguish the mentioned colors.
The figure and caption have been edited and we removed the MCMC chain.

Figure 6 caption: There seems to be some remains of previous versions of this figure in the caption? At least, I do not find the mentioned annual layer thickness profile in the figure.

Thank you. The caption has now been edited.

---

## Referee Report (RR1)

Re-Review of Muschitiello & Aquino-Lopez CPD

I'd like to thank the authors for their replies to my comments, which cleared up some of my questions but partly also reinforce some of my concerns. The additional analyses lend some support to the robustness of the results, yet, in the current version of the manuscript it is not clear how the synchronization is done exactly and whether it may be prone to biases.

A part of my confusion is stemming from the response of the authors to my previous comment on the underlying assumption of a linear relationship between speleothem and ice core data. I remarked, that equation 3 requires a linear relationship between the proxies. The authors argue in their response that this is overcome by standardizing the data to [-1,1] and assigning large uncertainties to the signal. However, the applied changes in the manuscript (L217-221) only outline that a non-linear adjustment of the timescales is facilitated by the method. This is of course obvious (and could be removed from the manuscript) but does not address my original question.

From the author's reply, I understand (but I am not sure because the manuscript and the reply are incoherent in this respect) that the standardization of the data is done separately for each 180-year segment. If so, this is not clearly stated in the manuscript (compare L253-257). Further, this raises additional questions.

1. A standardization for each segment would allow drastic changes in the relationship (regression slope) between ice core and speleothem data. It could lead to the large peak in the speleothem d18O data around 18-20 kaBP, which has the magnitude of a DO-transition, to be matched to some minor structure in the ice core record, which has no equivalent change there. What would be the reason for such a drastic change in the relationship? How valid is the assumption that this still reflects the same physical driver in both proxies? Allowing for a freely varying relationship between the proxies increases the likelihood of erroneously aligning signals that have no physical connection. I also disagree, that the standardization is better at handling non-linearity in the relationship between the proxies, since the full dynamic range of the proxies occurs at DO-onsets within decades which fits inside one standardization-segment and is thus, still treated as a linear relationship. Further, a standardization of each 180-year segment effectively corresponds to a 180-year high-pass filter, while the speleothem data has very little variability in this frequency band, especially during MIS-3.

2. A standardization of the record as a whole (as shown in the figures 4-6) on the other hand, leads to long periods of systematic differences between the records (MIS-2 but also during stadials of MIS-3). Because the method evaluates squared differences between the records (equation 3) this leaves the method prone to minimizing those differences instead of matching structures. To test this, I ran a test on artificial data (figure below), which are composed of a AR(1)-process (the exact same in both datasets) and different linear trends in both datasets, as seen in the real data. Standardizing both datasets (as a whole) to [1,1] and allowing for a linear timescale compression/stretching of one dataset by +/- 5% clearly leads to an erroneous compression of the timescale for the investigated segment. Increasing the uncertainty of the records does not fix this problem in contrast to the statement in the manuscript (L253-257). I understand that this would obviously be different when all segments of the data are jointly evaluated in the MCMC, but it is not clear whether this really avoids the problem as a whole. This effect may for example bias the inference during MIS-3 since the scaling leads to large difference between the records during stadials and small differences during interstadials (see figure 4) and I wonder whether this can explain part of the difference to the results by Corrick et al. (see also comment below).

This leads to several request for clarification/revision in the manuscript:

1. Please clearly indicate whether the standardization is done for each segment or for the record as a whole.
   a. If done for each segment: Please include a supplementary figure where you show the records after standardization of each segment. As it is now, the upper two panels of figures 4-6 are misleading. Furthermore, please discuss (incl. figure in SI) how variable the ratio of the scaling factors is over time (i.e., how variable is the assumed relationship between the proxies) and whether it can still be assumed that this reflects a common climatic process in both proxies.
   b. If done for the record as a whole, please show that the issue outlined under point 2 above is not affecting the synchronization (for example by analysing subsections of the data and standardizing those).
2. In both cases, I repeat my request for additional panels in figures 4-6 that show the correlation per segment before and after synchronization, to allow the reader to evaluate which signals are really driving the synchronization, and by how much the fit between the records is improved by synchronization. It is for example surprising that the synchronizations for the different datasets (NGRIP d18O, GIRP d18O, NGRIP Ca) is so similar in MIS-2 when these records have been shown to diverge during this period (see figure 2 but also Rasmussen et al. 2008 fig. 3, 10.1016/j.quascirev.2007.01.016). I am aware that it is not the correlation of records that is being evaluated, but it is an intuitive measure for the readers, and ultimately, a correlation of the signals is the fundamental reason why climate-wiggle matching is considered a valid tool in paleoclimatology. Further, this would illustrate how "continuous" the synchronization really is and where the transfer function is driven by the priors. If the synchronization is hinging on relatively few tie-points, then the title and main text need to be adjusted accordingly.

Another aspect that may need to be revised is the inclusion of the results by Corrick et al. Comparing the way how the authors present the results by Corrick et al. in figure 7 to the equivalent plot in the original publication (Corrick et al. figure S6) it appears that the authors exaggerate the uncertainties by Corrick et al. It is my impression, that they included the GICC05 uncertainty into this figure, which is irrelevant for this comparison. If this is done correctly, it becomes apparent that the results presented here, significantly disagree with those by Corrick et al. between 30-38 kaBP. This should be discussed as this is also the period where there seems to be a systematic disagreement to the match by Buizert et al. 2015, which the authors attribute to Buizert et al's use of Hulu-cave only. However, this argument would not hold for the results by Corrick et al. Further, as the authors state in L365ff, their synchronization of this period is largely driven by DO-onsets, which is similar to the estimates by Corrick et al. This difference may stem from a bias mentioned above (small differences between the records during interstadials, large differences during stadials). Alternatively, this disagreement may arise from the method trying to find a compromise between aligning GS-GI and GI-GS transitions, while not violating the counting error constraints? Please discuss. Again, additional panels with running correlations of similar would help evaluating this.

Specific Comments:

L71: "14C concentrations" – change to D14C which is not a concentration

L72: "ocean carbon inventories" – change to "radiocarbon inventories"

L86: Please include Adolphi et al. 2018 into the reference list, since we the main point of our work was to test the synchroneity of DO-events in speleothems and ice cores.

L110-114: These issues have nothing to do with the autocorrelation of cosmogenic radionuclides (d18O and Ca are autocorrelated as well) but are an artefact of analysing overlapping windows. Rephrase or delete.

L121: "when timescales reach their largest offset" – please add "according to cosmogenic radionuclides (Adolphi et al. 2018, Sinnl et al. 2023)"

L123: "first continuous" – previous transfer functions where also continuous, albeit based on selected tie-points and various ways to interpolate in between. Given that this method is likely also only driven by a limited number of tie-points it is not that different. Please adjust the formulation and possibly the title.

L129: "improve precision and accuracy" – how do you determine that your transfer is more accurate than previous versions? Please elaborate or delete.

L135: "three independent synchronization" – synchronization should be plural. However, change to "three synchronizations based on independent Greenland ice core climate proxy records" or similar. The synchronizations are not independent (always the same speleothem data).

L253-257: See major comments. Is the standardization done per segment? If yes, please clearly indicate.

L265 (eq3): There is an error in this equation. The power of two only applies to the numerator of the last term of the equation (squared differences).

L269: "Any underestimation" – of what?

L277: See my previous comments. Shouldn't tau0 be constrained by the MCE instead of the RCE?

L296: replace "uncertainties" with "credible intervals"

L346: Muscheler et al. 2008 inferred an offset of 65 not 55 years. Please change.

---

## Author Response (AR2)

We thank both Reviewers for their supportive reviews and the constructive comments. As R2 is happy with our previous revision, here we will only address R1's comments.
######################################################################################

Re-Review of Muschitiello & Aquino-Lopez CPD

I'd like to thank the authors for their replies to my comments, which cleared up some of my questions but partly also reinforce some of my concerns. The additional analyses lend some support to the robustness of the results, yet, in the current version of the manuscript it is not clear how the synchronization is done exactly and whether it may be prone to biases.

Thank you for your detailed feedback. We appreciate the reviewers time and effort in improving the current manuscript. We appreciate the opportunity to clarify our methodology and the rationale behind the standardization of the data.

A part of my confusion is stemming from the response of the authors to my previous comment on the underlying assumption of a linear relationship between speleothem and ice core data. I remarked, that equation 3 requires a linear relationship between the proxies. The authors argue in their response that this is overcome by standardizing the data to [-1,1] and assigning large uncertainties to the signal. However, the applied changes in the manuscript (L217-221) only outline that a non-linear adjustment of the timescales is facilitated by the method. This is of course obvious (and could be removed from the manuscript) but does not address my original question.

Apologies. There was some confusion around the reviewer's original question. We now understand the concern regarding the implication of assuming a linear relationship between the proxies as interpreted by Equation 3. In response, we would like to emphasize that the standardization of proxy data to a range of [-1, 1] and the assignment of large uncertainties to the signal are methodological choices aimed at mitigating the impact of assuming a strict linear relationship. The standardization process is not merely a procedural step but a crucial approach to minimize parameter estimation errors. On the other hand, the synchronization is performed in the time window of the target core ($t \in (t'_0, t'_m)$), allowing the method to put both records on a similar window, in both axis, in order to identify the best alignment that maximizes their similarity, as stated by the equation $u(t_i) \approx g(\tau(t_i)) \, \forall \, t_i \in \vec{t}$ (new lines 245). We would like to reinforce the point that this function seeks to identify similarities and is not to be taken as an equality of the point estimates, as stated by the use of the approximation symbol ($\approx$) and not the equality symbol (=).

It is important to note that this approach fundamentally differs from subjective point estimation of climate-wiggle matching. Our methodology relies on the uncertainty quantification of the whole record, and the equation u(t') ≈ g(τ(t')), together with the use of the t-distribution and conservative standard deviation, provides a conservative estimate for quantifying the uncertainty around the alignment between the records, resulting in a "best" guess for the alignment between both records, rather than subjective matching.

From the author's reply, I understand (but I am not sure because the manuscript and the reply are incoherent in this respect) that the standardization of the data is done separately for each 180-year segment. If so, this is not clearly stated in the manuscript (compare L253-257). Further, this raises additional questions.

1. A standardization for each segment would allow drastic changes in the relationship (regression slope) between ice core and speleothem data. It could lead to the large peak in the speleothem d18O data around 18-20 kaBP, which has the magnitude of a DO-transition, to be matched to some minor structure in the ice core record, which has no equivalent change there. What would be the reason for such a drastic change in the relationship? How valid is the assumption that this still reflects the same physical driver in both proxies? Allowing for a freely varying relationship between the proxies increases the likelihood of erroneously aligning signals that have no physical connection. I also

disagree, that the standardization is better at handling non-linearity in the relationship between the proxies, since the full dynamic range of the proxies occurs at DO-onsets within decades which fits inside one standardization-segment and is thus, still treated as a linear relationship. Further, a standardization of each 180-year segment effectively corresponds to a 180-year high-pass filter, while the speleothem data has very little variability in this frequency band, especially during MIS-3.

2.  A standardization of the record as a whole (as shown in the figures 4-6) on the other hand, leads to long periods of systematic differences between the records (MIS-2 but also during stadials of MIS-3). Because the method evaluates squared differences between the records (equation 3) this leaves the method prone to minimizing those differences instead of matching structures. To test this, I ran a test on artificial data (figure below), which are composed of a AR(1)-process (the exact same in both datasets) and different linear trends in both datasets, as seen in the real data. Standardizing both datasets (as a whole) to [1,1] and allowing for a linear timescale compression/stretching of one dataset by +/- 5% clearly leads to an erroneous compression of the timescale for the investigated segment. Increasing the uncertainty of the records does not fix this problem in contrast to the statement in the manuscript (L253-257). I understand that this would obviously be different when all segments of the data are jointly evaluated in the MCMC, but it is not clear whether this really avoids the problem as a whole. This effect may for example bias the inference during MIS-3 since the scaling leads to large difference between the records during stadials and small differences during interstadials (see figure 4) and I wonder whether this can explain part of the difference to the results by Corrick et al. (see also comment below).

Thank you for raising these additional points regarding the potential issues with standardizing the entire record and the impact of systematic differences between the records during certain periods. We now clearly state in the main text that the scaling is performed on the entire timeseries and not by segment (new lines 247 and 256). In the following we will therefore address point 2 from above.

Regarding the comment about standardizing the entire record potentially leading to long periods of systematic differences between the records, and the method minimizing these differences instead of matching structures, we acknowledge this as a valid concern. However, it is important to note that our methodology quantifies the uncertainty associated with the global alignment itself, evaluating the alignment quality across the entire record. This means that if there is a mismatch between the records in a particular period, it would also affect the sections where the records are properly aligned. While the standardization of the entire record may introduce biases during certain periods, our approach relies on the joint evaluation of all segments in the MCMC process. This global evaluation aims to find the optimal alignment that minimizes the overall differences while accounting for the uncertainties across the entire record. In other words, our methodology provides a conservative estimate of the uncertainty around the alignment by jointly considering all segments and their respective uncertainties. In addition, the use of the t-distribution and conservative standard deviation further contributes to this conservative estimate. We now more clearly discuss all these issues in the main text (new lines 256-261 and lines 345-353).

We appreciate the reviewer's point regarding the potential issues that may arise in different windows of observation. To test the robustness of our methods, we performed sensitivity tests where we align NGRIP $\delta^{18}O$ against the speleothem stack using short segments of ~10 kyr. We cropped both the input and target data and standardized the timeseries in the same way as per the global alignment. The sensitivity tests show that both alignments agree with each other, and more importantly, the differences between the global and local alignments are statistically indistinguishable (see figure below). The overlapping credible intervals across most of the record demonstrate the robustness of our approach, even when considering localized alignments. These new findings are presented and discussed in the new version of the manuscript (new Supplementary Fig. 1 and new lines 345-353).

[Figure]

**Supplementary Figure 1.** Comparison between the global and localized synchronizations of NGRIP $\delta^{18}O$ and EASM PC1. The records were cropped in segments of approximately 10 kyr and scaled between -1 and 1 before alignment. Intervals spanning 11-22 kyr b2k (**a**), 18-30 kyr b2k (**b**), 28-41 kyr b2k (**c**), and 38-48 kyr b2k (**d**). The target is always longer than the input by allowing 2 kyr on both sides of the timeseries. **e.** Posterior median and pointwise 95% credible intervals (of the difference $\Delta T$ between the GICC05 and U-Th timescales estimated locally (coloured lines) and globally (grey shading and black line).

As to the reviewer's concern about the potential bias during MIS-3, where the scaling may lead to large differences during stadials and small differences during interstadials, we now show that our results are in better agreement with Buizert et al's Δt estimates than previously thought after that the Hulu data is placed on the updated U-Th timescale (new Figs. 4-6, new line 359, and reply below).

This leads to several request for clarification/revision in the manuscript:

1. Please clearly indicate whether the standardization is done for each segment or for the record as a whole.

a. If done for each segment: Please include a supplementary figure where you show the records after standardization of each segment. As it is now, the upper two panels of figures 4-6 are misleading. Furthermore, please discuss (incl. figure in SI) how variable the ratio of the scaling factors is over time (i.e., how variable is the assumed relationship between the proxies) and whether it can still be assumed that this reflects a common climatic process in both proxies.

The standardization is performed for the entire record, not for each individual segment. We have clarified this in the revised text (new lines 245 and 256) and crafted a new Supplementary Figure to

compare the quality of the localized and global synchronizations. We agree that this process may struggle in records with extreme maxima/minima. We have added a discussion of this potential limitation in the main text (new lines 256-261 and 345-354).

b.  If done for the record as a whole, please show that the issue outlined under point 2 above is not affecting the synchronization (for example by analysing subsections of the data and standardizing those).

    We have address this in the points above and have also analyzed subsections of the data (see Supplementary Figure 1).

2.  In both cases, I repeat my request for additional panels in figures 4-6 that show the correlation per segment before and after synchronization, to allow the reader to evaluate which signals are really driving the synchronization, and by how much the fit between the records is improved by synchronization. It is for example surprising that the synchronizations for the different datasets (NGRIP d18O, GIRP d18O, NGRIP Ca) is so similar in MIS-2 when these records have been shown to diverge during this period (see figure 2 but also Rasmussen et al. 2008 fig. 3, 10.1016/j.quascirev.2007.01.016). I am aware that it is not the correlation of records that is being evaluated, but it is an intuitive measure for the readers, and ultimately, a correlation of the signals is the fundamental reason why climate-wiggle matching is considered a valid tool in paleoclimatology. Further, this would illustrate how "continuous" the synchronization really is and where the transfer function is driven by the priors. If the synchronization is hinging on relatively few tie-points, then the title and main text need to be adjusted accordingly.

    We disagree that the three synchronizations are similar during MIS2. Perhaps the trends/trajectories are comparable but the resulting offsets are quantitatively different and involve uncertainties of at least 0.1kyr in either direction. We now show the extent of the credible intervals associated with each synchronization (panels d of new Figures 4-6). These intervals will aid the reader identifying regions where the alignment is less robust. From these plots, it is evident that there are no sudden jumps in the synchronization uncertainty, nor a systematic narrowing of the uncertainty at the onset of stadials and interstadials, thus demonstrating that the alignment is not driven by discrete "tie points". Instead, the synchronizations yield relatively smooth uncertainty bounds throughout, which supports our argument of a continuous synchronization. This approach –which aligns with the Bayesian methodology– should satisfy the reviewers interest and provides the same intuitive measure for the reader.

    Regarding the role of the priors, the figure below shows the prior versus posterior distributions for the RCE parameters. The interval widths demonstrate that the posteriors differ substantially from the priors, thus confirming that the data is driving the synchronization estimates, not just the priors, as the reviewer suggested.

[Figure]

Prior and posterior RCE parameters for each CLIM synchronization.

Another aspect that may need to be revised is the inclusion of the results by Corrick et al. Comparing the way how the authors present the results by Corrick et al. in figure 7 to the equivalent plot in the original publication (Corrick et al. figure S6) it appears that the authors exaggerate the uncertainties by Corrick et al. It is my impression, that they included the GICC05 uncertainty into this figure, which is irrelevant for this comparison. If this is done correctly, it becomes apparent that the results presented here, significantly disagree with those by Corrick et al. between 30-38 kaBP. This should be discussed as this is also the period where there seems to be a systematic disagreement to the match by Buizert et al. 2015, which the authors attribute to Buizert et al's use of Hulu-cave only. However, this argument would not hold for the results by Corrick et al. Further, as the authors state in L365ff, their synchronization of this period is largely driven by DO-onsets, which is similar to the estimates by Corrick et al. This difference may stem from a bias mentioned above (small differences between the records during interstadials, large differences during stadials). Alternatively, this disagreement may arise from the method trying to find a compromise between aligning GS-GI and GI-GS transitions, while not violating the counting error constraints? Please discuss. Again, additional panels with running correlations of similar would help evaluating this.

Thank you for bringing this up. We took this opportunity to revise Buizert et al's match points to

account for the new U-Th chronology presented in Cheng et al., 2018 and to assess whether said match points still hold. To update Buizert et al's Δt estimates we adopted a simple sliding correlation approach using windows of 2000 years around the published tie points that are moved in steps of 10 years with a maximum lead/lag of 500 years (see figure below and Δt corrections on panels c of new Figures 4-6). The corrected Δt estimates are within the 95% credible intervals of our transfer functions and reveal a possible scaling of GICC05 that is slightly larger than previously estimated (0.99%).

[Figure]

Comparison between the speleothem $\delta^{18}O$ stack from Cheng et al. (2016) and the Hulu Cave $\delta^{18}O$ data from Cheng et al. (2018) on the timescale published therein. Black squares and horizontal bars indicate the age shift due to the new chronology at the interstadial transitions identified by Buizert et al. (2015). Positive values imply that the new timescale is older than the previous chronology. Age shifts were estimated using cross-correlation, whereby we correlated windows of 2000 years length with leads/lags of 500 years at steps of 10 years centered around said tie points.

As to comparing our results with the Δt estimates presented in Corrick et al. (2020), we adjusted the uncertainty bars as requested by the reviewer (see new Fig. 7). In general, we are happy to include additional text and discuss further. However, we deem such estimates to be –to some extent– subjective, and therefore inconsistent with our methodology. We strongly suggest that their age offsets should not be over-interpreted and in fact taken with a grain of salt. Admittedly, the authors state that their results differ from Buizert et al's Δt estimates as well as results from cosmogenic radionuclide wiggle matching, and that such difference is potentially associated with their "*methodological approach, including the choice of detrital-thorium correction and depth-age modeling*". More critically, the identification of interstadials is rather qualitative, and reliant on the visual identification of "*the first data point of the steep part that clearly deviates from the baseline level preceding the transition*" using at times low-resolution records that have "*at least three data points per thousand years*". In addition, the authors state that "*it was occasionally necessary to shift the point to a position structurally similar to that of the event's assigned position in NGRIP*". As such, they state that "*statistical methods to identify the onset of interstadial transitions were found to be difficult to implement consistently to all speleothem records*". The qualitative and subjective nature of Corrick et al's approach is at odds with the more objective method presented in our study and therefore differences in the resulting Δt estimates are not surprising. As a final remark, it should be noted that the authors analyzed the Hulu Cave data using the "pre-2018" U-Th chronology (i.e. Wang et al., 2001; Wu et al., 2009; Southon et al., 2012), which may have additionally contributed to the observed mismatch.

We now discuss more openly these issues in the main text (new lines 396-404).

Specific Comments:

L71: "14C concentrations" – change to D14C which is not a concentration L72: "ocean carbon inventories" – change to "radiocarbon inventories"

Thank you. This is has been corrected (new lines 71-72).

L86: Please include Adolphi et al. 2018 into the reference list, since we the main point of our work was to test the synchroneity of DO-events in speleothems and ice cores.

This has been added (new line 85).

L110-114: These issues have nothing to do with the autocorrelation of cosmogenic radionuclides (d18O and Ca are autocorrelated as well) but are an artefact of analysing overlapping windows. Rephrase or delete.

This has been deleted and rephrased (new line 111).

L121: "when timescales reach their largest offset" – please add "according to cosmogenic radionuclides (Adolphi et al. 2018, Sinnl et al. 2023)"

This has been added (new line 121).

L123: "first continuous" – previous transfer functions where also continuous, albeit based on selected tie-points and various ways to interpolate in between. Given that this method is likely also only driven by a limited number of tie-points it is not that different. Please adjust the formulation and possibly the title.

While previous transfer functions were indeed continuous, they relied on (at times subjectively) selected tie points and various interpolation techniques between those points. In contrast, our approach is not driven by pre-selected tie points but rather a quantifiable assumption of continuity across the entire record. Furthermore we provide an objective uncertainty quantification of the overall alignment between the two records, rather than relying on interpolation between sparse tie-points. This objective uncertainty quantification is a key distinguishing feature of our method, as it allows evaluating the robustness of the alignment across the entire input. With this in mind we believe that no change is needed.

L129: "improve precision and accuracy" – how do you determine that your transfer is more accurate than previous versions? Please elaborate or delete.

This has been deleted (new line 130).

L135: "three independent synchronization" – synchronization should be plural. However, change to "three synchronizations based on independent Greenland ice core climate proxy records" or similar. The synchronizations are not independent (always the same speleothem data).

Thank you. This has been changed accordingly (new lines 136-137) and the term "independent" has been removed throughout the text.

L253-257: See major comments. Is the standardization done per segment? If yes, please clearly indicate.

Thank you. This has now been clarified in the main text. Please see our detailed reply above.

L265 (eq3): There is an error in this equation. The power of two only applies to the numerator of the last term of the equation (squared differences). old

The reviewer is correct we had made a small typo in the equation. This has now been corrected in the main text (new Equation 3):

$$l \propto \sum_{i=0}^{n} \left[ -\log(\sigma_{u_i}) - \frac{a}{2} \log\left( b + \frac{(G(t_i) - u_i)^2}{2\sigma_{u_i}^2} \right) \right],$$

L269: "Any underestimation" – of what?

This has been rephrased (new lines 277-278).

L277: See my previous comments. Shouldn't tau0 be constrained by the MCE instead of the RCE?

Thank you. This section has been edited accordingly (new lines 286-292).

L296: replace "uncertainties" with "credible intervals"

This has been changed (new line 306).

L346: Muscheler et al. 2008 inferred an offset of 65 not 55 years. Please change.

Thank you. This has been changed (new line 364).